# The impacts of dust aerosol and convective available potential energy on precipitation vertical structure in southeastern China as seen from multiple source observations

Hongxia Zhu[1], Rui Li [1,2,3], Shuping Yang[1], Chun Zhao[1], Zhe Jiang[1], and Chen Huang[1]

[1]School of Earth and Space Science, Comparative Planetary Excellence Innovation Center, University of Science and Technology of China, Hefei 230026, China

[2]State Key Laboratory of Fire Science, University of Science and Technology of China, Hefei 230026, China

[3]Institut de recherche sur les forêts, Université du Québec en Abitibi-Témiscamingue (UQAT), Rouyn-Noranda, J9X 5E4, Canada

*Correspondence to*: Rui, Li (rli7@ustc.edu.cn)

**Abstract.** The potential impacts of dust aerosol and atmospheric convective available potential energy (CAPE) on the vertical development of precipitating clouds in southeastern China (110° E-125° E; 20° N-30° N) in June, July, and August during 2000 to 2013 were studied using multiple-sources observations. In southeastern China, heavy dusty conditions is coupled with strong north wind which carried airmass containing high concentration of mineral dust particles with cold temperature and strong wind shear. This leads to weaker CAPE in dusty days comparing with that of pristine days. Based on satellite observations, the precipitating drops under dusty conditions grow faster at middle layer (with temperature -5 °C to +2 °C) but slower at upper and lower layer comparing with the pristine counterpart. For a given precipitation top height (PTH), the precipitation rate under dusty conditions is weaker in upper layer but heavier at middle and lower layer. And the associated latent heating rate released by precipitation at middle layer is stronger. The precipitation top temperature (PTT) shows fairly good linear relationship with near surface rain rate (NSRR). The linear regression slope between PTT and NSRR are stable at dusty and pristine conditions. However, the $PTT_0$ (PTT related to rain onset) at the onset of rain are highly affected by both CAPE and aerosol conditions. In pristine days, stronger CAPE facilitates the vertical development of precipitation and leads to a decrease of $PTT_0$ at the rate of -0.65 °C per 100 J kg$^{-1}$ CAPE for deep convective precipitation with variation of 15 %, and by -0.41 °C per 100 J kg$^{-1}$ CAPE for stratiform precipitation with variation of 12 %. After removing the impacts of CAPE on PTT, dust aerosols lead to

an increase of $PTT_0$ at the rate by +4.19 °C per unit aerosol optical thickness (AOD) for deep convective

precipitation and by +0.35 °C per unit AOD for stratiform precipitation. This study showed clear

evidence that meteorology conditions are combined with aerosol conditions together to affects the

vertical development of precipitation clouds. And quantitative estimation of the sensitivity of PTT to

CAPE and dust were also provided.

**1    Introduction**

Dust aerosols are widely distributed in the troposphere, which can scatter and absorb solar shortwave

radiation and terrestrial longwave radiation thereby directly affecting the global radiation budget

(Bellouin et al., 2005; Huang et al., 2014). On the other hand, dust aerosols can act as ice nuclei (IN,

Demott et al., 2003; Atkinson et al., 2013; Li et al., 2017a) to enhance heterogeneous freezing process

which leads to ice formation at relatively higher temperature and lower vapor saturation ratio. In addition,

dust particles coated with water soluble pollutants can serve as cloud condensation nuclei (CCN, Yin and

Chen, 2007; Li et al., 2010) to decrease effective radius of cloud droplets for given liquid water content

and indirectly modulate warm rain process. The mechanisms of dust aerosol affecting atmospheric

hydrometers are distinct at different temperatures and altitudes.

Observational studies of dust aerosols affecting clouds and precipitation at different vertical layers have

received increasing attentions in recent years. In particular, the observation of the vertical structure of

precipitation by spaceborne precipitation radar make it possible to investigate the impacts of dynamics

and aerosols on the formation of cloud and rain at different heights and temperatures in detail.

Previous studies have shown that the vertical structure of precipitation is influenced by aerosols in

addition to atmospheric dynamic (Min et al., 2009; Li and Min, 2010; Fan et al., 2013, 2018; Rosenfeld

et al., 2014; Gibbons et al., 2018; Chen et al., 2016; Guo et al., 2018; Wall et al., 2015). In a case study

of interaction between Sahara dust and mesoscale convective system over equatorial Atlantic Ocean, Min

et al. (2009) found that convective precipitation rate in dust laden sector was weaker than that in pristine

environment; and the radar echoes of stratiform precipitation influenced by dust were stronger at upper

levels than those in pristine environment. Li and Min (2010) further analyzed the variation of

precipitation rate with height and found the impacts of mineral dust on tropical clouds and precipitation

systems are highly dependent on rain type. The convective rain rate weakened at all heights, but

stratiform precipitation showed enhanced rain rate above 6 km in dust laden area, indicating that dust aerosols enhanced the heterogeneous ice nucleation process. In their study, variations of precipitation

related to meteorology conditions were constrained by fixed rain type, the precipitation top height, etc. Gibbons et al. (2018) used cloud resolving model simulations to reveal that more and smaller ice particles release more latent heat (LH) during deposition growth and riming after being affected by dust, therefore promotes convective development. During diffusional growth, more particles competing for available water vapor reduces the particle growth rate, shifting the height of precipitation formation to higher

heights during heterogeneous nucleation regime. In addition, Guo et al. (2018) found that convective precipitation in polluted conditions had deeper and stronger radar reflectivity patterns, while stratiform and shallow precipitation had shallower and weaker patterns than those under pristine conditions. Observational and model simulation studies have shown different results for aerosol effects on deep convection, suggesting that aerosols may either invigorate or inhibit precipitation, depending on the type

and concentration of aerosols and environmental conditions (Jiang et al., 2018; Khain 2009; Fan et al., 2009, 2013; Rosenfeld et al., 2008, 2014).

The precipitation top height/temperature (PTH/PTT) is one of the most important parameters to represent the vertical structure of precipitation. It is mainly controlled by the strength of the updraft (Nasuno and Satoh, 2011). In addition, the colder PTT (the higher PTH) the raining system is, the longer falling path

of precipitation drops and stronger rain rate at the surface can be reached (Cao and Qi, 2014; Liu and Fu, 2001). Li et al. (2011b) found that for mixed-phase clouds, the cloud top height increases with the concentration of condensation nuclei, for water clouds, the cloud top height is insensitive to condensation nuclei. Dong et al. (2018) studied the effect of Sahara dust aerosols on PTH in the equatorial Atlantic Ocean, and found for a given near surface rain rate (NSRR), the PTH of stratiform precipitation in dusty

conditions was significantly higher than its pristine counterpart. In that study, it was found that the variations of rain fall vertical structure was dominated by dynamics which can explain about 90 % variance. Guo et al. (2018) found that the mean top heights of 30 dBZ radar reflectivity of polluted convective (stratiform) precipitation increased (decreased) by ~29% (~10.8%) comparing to that under pristine conditions.

The spatial distribution of aerosols was significantly affected by meteorological conditions (Oshima et al., 2012), including convective transport (Prospero and Mayol-Bracero, 2013), wet removal processes

(Park and Allen, 2015), boundary layer height and evolution processes (Li et al., 2017b). Aerosol-cloud-precipitation interactions (ACIs) also largely depend on meteorology conditions including wind shear (Fan et al., 2009, 2013), atmospheric stability (Huang et al., 2014), relative humidity (Li et al., 2019b), and the altitudes of the aerosol layer (Yin et al., 2012, Lee et al., 2022).

A great challenge in observational study on the indirect effects of aerosols is to distinguish the isolated contributions of weather conditions (dynamic conditions) and aerosol microphysical effects to the observed macro-micro features of clouds and precipitation (Stevens and Feingold 2009; Tao et al., 2012; Rosenfeld et al., 2014; Li et al., 2017). This is especially true for mesoscale convective systems (MCSs) that are heavily affected by large-scale atmospheric circulation. Some studies have adopted this ideals to constrain the variations of dynamical factors, cloud type, stages of cloud precipitation development and etc., and then to analyze the influence of aerosols (Rosenfeld et al.,2008; Fan et al.,2013, 2018; Li et al., 2011b; Min et al.,2009; Li and Min,2010; Gibbons et al., 2018). For example, Fan et al. (2013) found that the thermodynamic effect of aerosols (freezing of cloud water to release additional LH) contributes up to 27 % to the increase in cloud cover during the growth stage of deep convective clouds in summer, while the microphysical effect of aerosols (freezing of large amounts of cloud droplets to produce more and smaller ice particles) increases cloud cover and cloud top height during the mature and dissipation stages.

For areas far from the source of a certain type of aerosol (such as mineral dust), the occurrence of high aerosol concentrations is often accompanied by specific atmospheric circulation conditions (for long-distance transport of aerosols), then under this circumstances, the observed cloud and precipitation characteristics are jointly determined by the combination of obviously different aerosol conditions and weather conditions, if we want to understand the pure indirect effects of aerosols, we will have to untangle these two different effects.

To this end, this study specially selected southeast China as the research area. It is relatively far from the original source of dust, so a relatively fixed atmospheric circulation conditions (northwest wind) is required to transport dust to this area, which creates an ideal test bed for us to investigate combined effects from dust aerosol and meteorology conditions on precipitation. And we attempt to isolate the impacts from meteorology conditions and aerosol conditions on the vertical structure of precipitation and LH by analyzing multiple satellite observation with new mathematic treatment. Particularly, this study

investigates the effects of convective available potential energy (CAPE) and dust aerosol on the processes of precipitation particle formation, LH releasing, and the macrophysical vertical features of precipitation in the southeastern China. In addition, the sensitivity of PTT to CAPE and aerosol optical depth were quantitatively studied.

**2 Data and Method**

The southeastern China (110° E-125° E; 20° N-30° N) was selected as the study area. It is featured by strong and frequent summer precipitation and is far from the terrestrial dust original source. Therefore, the potential dust-precipitation interaction must occur under special weather conditions. This is useful for us to investigate the separated effects from aerosol and thermodynamics on clouds. And the study focused on precipitation in June, July and August (JJA) during 2000 to 2013.

The standard product 2A25 of Tropical Rainfall Measuring Mission (TRMM) was utilized in this study (Iguchi et al., 2000) with 4.3-km horizontal and 250-m vertical resolutions at nadir. From top to down, the height of the first three continuous vertical bins with PR detectable echoes are defined as precipitation top. And the associated air temperature (see below data collocation method) are defined as PTT. Each precipitation profile has a certain rain type which is either convective, stratiform or others (Awaka et al., 1997). Because the dynamic conditions in convective and stratiform precipitation are different (Houze, 1997), the potential dust related effects are also distinct (Li et al., 2017a; Li and Min, 2010), those two types of precipitation are investigated separately in this study. In addition, warm rain were separated from these two types. Warm rain is defined as those with PTT warmer than 0 °C.

Based on satellite radar observations (Liu and Fu, 2001), from the rain top to the surface, the precipitation rate (R, unit mm h$^{-1}$) at logarithm (i.e. logR) changes linearly with the decreasing height (H) in three vertical layers respectively. At the highest level, precipitation particle growth relies primarily on the water vapor deposition process, the growth rate is slow, and the linear regression slope of the logR to H is small. In the layer of about 1.5-2 km above and below the freezing level, precipitation particles rely on the process of aggregation, and riming to grow rapidly, and the corresponding linear regression slope is large. In the lower layer, the convective precipitation rate shows a further slight increase due to coalescence with cloud droplets end up with a very small linear regression slope. The stratiform precipitation rate in this layer is basically no longer growing due to the lack of cloud droplets. Both convective and stratiform

precipitation rate may decrease towards ground due to particle break-up and evaporation. Li et al. (2011a)

found above phenomenon standing valid when changing the vertical coordinate from height (H) to

temperature (T). They further defined the associated logR~T linear regression slopes ($\partial$logR $\partial$T$^{-1}$) as 1)

SlopeA in the layer with temperatures colder than -5 °C, 2) SlopeB in the middle layer with temperatures

between -5 °C to 2 °C, and 3) SlopeC in the lowest layer with temperatures higher than 2 °C respectively.

It was found the three slopes respond to variations of atmospheric dynamics and thermodynamics related

to El Nino. In this study, we adopted the definition of Slopes A, B, C to study the growth rate at upper,

middle and lower layers.

The formation of precipitation is accompanied by the release or depletion of latent heat (LH), which

plays an important role in maintaining the global energy balance (Houze, 1997; Li et al., 2019a). The

standard TRMM 2A25-base LH products derived from Convective and Stratiform Heating algorithm

(CSH, Tao et al., 1993, 2010), Spectrum Latent Heating algorithm (SLH, Shige et al., 2004, 2007), and

the recently developed Vertical Profile Heating algorithm (VPH, Li et al., 2011a, 2019a) were used in

this study to investigate the possible impacts of aerosols on precipitation LH.

The standard aerosol product MOD04_3K aerosol optical thickness (AOD) from Moderate Resolution

Imaging Spectroradiometer (MODIS) on Terra satellite at horizontal resolution of 3 km were used in this

study. The retrieved AOD, the fine mode fraction (FMF) and the coarse mode AOD (CMAOD= AOD×(1-

FMF)) were combined to define dusty and pristine conditions. Because AOD is not available under

cloudy sky, for each 1×1 grid where precipitation was detected by TRMM PR, the averaged AOD and

CMAOD from the surrounding eight grids are assigned to this grid. If the AOD of all eight grids are

missing, then the precipitating grids AOD were recorded as missing, and such grids were excluded from

this study. Otherwise the averaged AOD from the 8 grids AOD is assigned to precipitating grid (it is not

required that all 8 grids have AOD observations). Then the mean CMAOD from all precipitating grids at

the same day were calculated. If the mean CMAOD is larger than 0.5, then the day was defined as "dusty

day". And all rain samples in that day were defined as polluted rains. If the mean total AOD is less than

0.2, the day was defined as pristine day, and all rain samples in that day was defined as pristine rains.

Under this classification criteria, for convective (stratiform) precipitation, over 83% (84%) precipitating

grids in pristine days showed total AOD lower than 0.2, and over 87% (79%) precipitating grids in dusty

days showed CMAOD heavier than 0.5. In another word, such method can represent the main feature of

aerosol condition and it has the advantage to show the large-scale atmospheric circulation as an "ensemble" comparing to the method of defining the aerosol condition for each precipitation grid separately.

The atmospheric thermodynamic conditions under pristine or dusty environments were derived from hourly ERA5 reanalysis data at horizontal resolution of 0.25°×0.25° (Hersbach et al., 2020). The parameters including air temperature (T), zonal wind field (U) and meridional wind field (V) at the upper (300 hPa), middle (500 hPa) and lower (750 hPa) level and Convective Avilable Potential Energy (CAPE) are investigated. For each TRMM PR detected raining pixel, the daily averaged ERA5 variables averaged from all grids ±0.5° surrounded it are assigned to it.

## 3 Results

### 3.1 The coupling between aerosol conditions and meteorology conditions

The selected study area is undergoing fast economic development during recent decades. In summer, the area generally was dominated by anthropogenic emission related fine mode aerosols (Fig. 1a, b, c--- total AOD, CMAOD and FMAOD in all days). However, still there are some days in which heavy dust aerosol in this area can be observed by satellite with high value of coarse mode AOD (Fig. 1d-f). During 2000-2013 JJA, there were 46 raining days that were defined as dusty conditions with CMAOD exceeded 0.5 averaged from raining pixels. Meanwhile, 92 raining days that were defined as pristine conditions with area mean total AOD smaller than 0.2.

The study area is not the original emission source of mineral dust, instead, the satellite observed dust was transported from remote desert areas. Special large scale atmospheric circulation conditions are required to show significant dust aerosol in this area. For example, on 12 June 2006 a typical dusty precipitation day, about half of the study area was covered by heavy dust (Fig. 2) with satellite observed CMAOD up to 1. The back trajectory analysis using HYSPLIT (Fig. 2) showed the dusty air mass were from the Gobi and/or Taklamakan Desert. We likewise examined the backward trajectories of other dusty days, such as 20 June 2010 (Fig. S1), 11 June 2012 (Fig. S2), 16 June 2012 (Fig. S3), etc., the dusty air mass was from the Gobi and/or Taklamakan Desert as well. This suggests that the backward trajectory of the dusty air mass on 12 June 2006 is representative for the whole study period. Liu et al. (2011) also found that dust in southeast China originated from the Gobi Desert and Taklamakan Desert in northwestern China.

Large scale circulations at 300, 500, and 750 hPa for dusty and pristine days were analyzed (Figs. 3 and S4). Generally, the dust emission source area in northern China is under the control of the westerly wind. When the selected study area was in dusty conditions, in 35 °N -50 °N belt the southward wind component was significantly strengthened, which transported the dust aerosol southwardly into the study area (Fig. S4). At the same time, it also transported colder air masses from the north to the south, resulting in temperature at 500 hPa about 1 degree colder than that in the pristine conditions. In addition, when the study area was in the dusty conditions, the 1000-500 hPa layer U wind shear was about 2 times of that in pristine conditions (5.1 m s$^{-1}$ vs. 2.9 m s$^{-1}$). Finally, as an overall measure of regional mean atmospheric insatiability, regional CAPE was 600 J kg$^{-1}$ in dust conditions and 743 J kg$^{-1}$ in pristine conditions.

In summary, in southeastern China, heavy dusty condition is generally accompanied by certain synoptic pattern dominated by strong north wind. The spatial correlation coefficient between CMAOD and CAPE is -0.07, and is -0.08 for CMAOD against T (700 hPa), 0.09 for CMAOD against U windshear, and 0.07 for U wind at 700 hPa respectively. Comparing the dusty conditions to the pristine conditions, its CAPE is lower, the air temperature is colder, the U wind shear is stronger and the north wind is enhanced. In both dusty and pristine precipitation days, the synoptic forcing conditions favor the lifting of air mass and convection initiation comparing to that in non-precipitating days. Statistically, pristine precipitation events are featured by relatively higher CAPE and lower wind shear conditions, which may enhance the vertical development of the precipitating clouds. In the following discussion, the CAPE were used as an overall indicator for investigating the impacts of meteorology conditions on precipitation features. It should be emphasized that the difference in CAPE should be mainly determined by synoptic conditions instead of aerosol.

**3.2 Differences in vertical profiles of precipitation**

The precipitation vertical profiles, i.e., the function of the precipitation rate changing with the vertical height (temperature), contains information of precipitating particles growing mechanisms and speed during the falling from precipitation top to the surface ( Liu and Fu, 2001). Li et al. (2011a) further discussed how these profiles can be affected by large-scale circulations, such as those in the different phases of ENSO. Li et al. (2019a) directly uses the vertical gradient of precipitation rate to estimate the latent heating rate in clouds. However, few study reported the possible effects of dust aerosols on the shape of precipitation profiles.

It is expected that changes in dynamic conditions will lead to changes in the precipitation profile. As shown in Fig. 4a-b, for a given NSRR, the PTT/PTH in pristine conditions (dotted curves) are colder (higher) than those in dusty conditions (solid curves) for deep stratiform and convective precipitation. In the layer from precipitation top to about -10 to -5 °C , the mean precipitation rates in pristine conditions are heavier than those under dusty conditions. However, in next layer when temperature is ranging from -5 °C to 2 °C, the precipitation rate in dusty conditions grows much faster than that in pristine conditions. The effect is so significant that the dusty precipitation rate exceeds pristine precipitation rate at about 0 °C and keep growing rapidly. In the lowest layer close to surface, precipitation rate under dusty conditions grows slower.

From another angle, when dropping from the same PTTs (Fig. 4d, e), the precipitating particles in dusty conditions grow slower than its pristine counterpart at upmost layer. Starting from temperature around -10 °C to -5 °C , the dusty precipitating particles grow faster and obtain large amount of water mass in the middle layer. Although followed by a layer with slower growing, the final NSRR for given PTT under dusty conditions (solid curve) is still heavier than that of pristine rain (dotted curve). Such effect is weak for stratiform rains particularly those with relatively warm PTTs (e.g. light blue and green curves in Fig. 4d). This is because the proposed dust's IN effect generally works for ice-phase microphysical process, for those stratiform rains start from warm PTTs, there is no sufficient water content and the temperatures are too warm to heterogeneous freezing take place.

For the warm rains without ice phase microphysical processes, for given NSRR, colder PTT is required for dusty rains (Fig. 4c). When dropping from the same PTT, dusty rain rate increases slower than pristine rain and is weaker at the near surface (Fig. 4f). This indicates a possible suppression by dusty conditions for warm rain growth. During the long-range transportation of dust from north to southeastern China, very likely the dust particles were coated by soluble aerosols and become active CCN (Li et al., 2010) in the warm rains. For given condensed liquid water content, this additional CCN leads to smaller cloud effective radius thus decreases the coalescence efficiency which is the main mechanism for warm rain growth (Rosenfeld et al., 2008; Min et al., 2009; Yin and Chen, 2007; Li et al., 2010).

Accompanied the changes of vertical profiles, the latent heating released from dusty precipitation are also changed comparing to the pristine precipitation. Figure 5 represents the contoured frequency by altitude diagrams (CFAD) of LH (retrieval from VPH method) for deep stratiform rains and convective

rains, respectively. Under dusty conditions, stratiform and convective rains exhibits an increased positive heating near 5 km altitude and a decrease of negative heating (cooling) at higher layer. From the difference of CFAD of LH (Fig. 5c and 5f), the negative values of the difference all appear around 5km where the heterogeneous freezing process dominates. The presence of dust intensifies the heterogeneous freezing process, making it easier for ice to form, resulting in an increase in positive heating and a decrease in cooling. The LH vertical structure of stratiform and convective rains have similar feedback to dust aerosol. Meanwhile, the cooling (i.e. negative LH) in layer lower than 5 km is also enhanced based on Fig. 5c and 5f.

Figure 6 shows the mean LH profiles for stratiform and convective precipitation derived from the three different LH algorithms (i.e. SLH, CSH and VPH). For stratiform rains, there is no significant difference between SLH and CSH between pristine and dusty precipitation at all heights, while VPH shows a stronger latent heat in the dusty conditions near 5-6 km.

For deep convective rains, VPH shows that the LH in the dusty conditions is weaker than that in the pristine environment in the upper layers above 8 km, while is stronger in the middle layers around 5-6 km. Both CSH and SLH showed neglectable differences at upper layer, but also showed stronger LH in dusty conditions in middle layer and lower layers comparing to pristine conditions.

There are significant differences of mean LH profiles among the three algorithms indicating large uncertainties in satellite retrieval of LH. However, all the three products agree that LH in deep convective precipitation at middle layer (around 5-6 km) in dusty conditions should be stronger than those in pristine conditions.

Validation of satellite retrieved LH is still a very challenging task (Tao et al., 2022) because there is no directly measured ground-truth of LH available. Intercomparison among different LH products is one of the useful indirect means to evaluate their accuracy. Based on Li et al., (2019a), VPH product showed reasonable structure of LH in Tibetan Plateau with similarities and dissimilarities comparing to CSH and SLH. In this study, the VPH product was chosen because it is directly related to the variations of precipitation rate at each altitude, while CSH and SLH retrievals use constrains of precipitation rate at surface, precipitation top height, precipitation type, etc. It should be emphasized, the LH-related results did not receive rigorous validation in this study area, thus should be treated with cautions.

**3.3 Differences in the growth rate of precipitation**

In this section we adopted the same method of Li et al. (2011a) to use the linear regression slope of precipitation rate (at logarithm) to temperature (i.e. dlogR dT$^{-1}$) to quantify the differences of growth rate

at different layers between dusty and pristine conditions.

In a NSRR-PTT space, the mean SlopeA (upper layers), SlopeB (middle layers), and SlopeC (lower layers) under pristine and dusty conditions were compared as shown in Figs. 7, 8 and S2.

For stratiform and convective precipitation (Figs. 7 and 8), for a given NSRR, all slopes decrease with decreasing PTT (i.e. increasing PTH). For a given PTT, SlopeA and SlopeB increase with NSRR, while

SlopeC is almost insensitive to NSRR. This indicates the growth rate at upper layer and middle layer are critical to determine the final surface rain rate. It should be noticed that even both PTT and NSRR are constrained, still SlopeB in dusty conditions is significantly stronger than that in pristine conditions (Figs. 7f and 8f), meanwhile, SlopeA is not significantly different, and SlopeC is weaker in dusty conditions than that in pristine conditions.

As for warm rain (Fig. S5), for a given NSRR, SlopeC increase with increasing PTT. For a given PTT, SlopeC increase with NSRR. Even both PTT and NSRR are constrained, still SlopeC in dusty conditions is significantly weaker than that in pristine conditions (Fig. S5c).

For a given PTT, mean SlopeA of pristine stratiform precipitation (dash curve) is slightly greater than that for dusty precipitation (Fig. S6a). However, Slope B for both convective and stratiform rains in dusty

conditions are remarkably greater than its pristine counterpart (Fig. S6c, d). And the t testing showed that most differences of SlopeB exceeded the 95 % or 99 % confidence level (Fig. S7). This finding strongly supports the hypothesis that dust aerosol enhanced the heterogeneous freezing process at temperature much higher than -38 °C (the threshold for homogeneous freezing). There are more ice phase hydrometeors in the middle layer of dusty conditions to favor the aggregation and rimming process so

that precipitation drops can grow faster than those in pristine conditions. And the slightly smaller Slope A for dusty conditions can be explained that the fast formation of cloud and precipitation in middle layer exhaust water vapor at middle layer, few water vapor at upper layer can be used for precipitation growth in dust conditions.

In the lowest layer, when precipitation particles fall to the ground, the SlopeC of stratiform precipitation

is basically negative (Fig. S6e), which corresponds to the evaporative and/or breakup process of raindrops. From the analysis in Section 3.1, it is known that the dusty conditions corresponds to a stronger U

windshear (Fig. 3d-f), and the stronger windshear may enhance the evaporative and/or breakup processes (Fan et al., 2009, 2013; Li et at., 2010). It was confirmed in Fig. S6 that the SlopeC of stratiform rains in dusty conditions is more negative than that in pristine conditions. For convective precipitation in the lowest layer, raindrops can still slightly increase through the coalescence of cloud droplets. Based on the observations, the SlopeC of convective rains in dusty conditions is smaller than pristine conditions, indicating the coalescence process of cloud droplets was suppressed. The strong windshear in the dusty conditions may cause this suppression. As for warm rain, the SlopeC in dusty conditions is significantly smaller than that in pristine conditions and the t testing showed that differences of SlopeC exceeded the 99 % confidence level (Fig. S8), indicating that dust suppress warm rain. In addition, polluted dust particles may also act as CCN to decrease the effective radius of cloud droplets and inhibit the coalescence efficiency (warm rain) as suggested by Rosenfeld (2008), Li et al. (2010), Min et al. (2009) and Yin and Chen (2007).

It is interesting that the dependence of Slope on PTT is getting stronger from C to A (Fig. S6). The precipitation particle growth rate at upper layer (water vapor deposition process) and middle layer (aggregation and riming process) are critical to determine the final surface rain rate. SlopeA and SlopeB are more sensitive to PTT. As for SlopeC in the lower layer, the convective precipitation rate has a slight increase due to coalescence with cloud droplets. However, in the layer very close to surface, rain rate no longer grows but decreases due to breakup and/or evaporation. For the stratiform precipitation, rain rate in this layer does not grow due to the lack of updraft. Therefore, SlopeC is not sensitive to PTT.

### 3.4 The sensitivity of PTT to CAPE and aerosol

PTT has a close relationship with surface rain rate (NSRR) because the higher PTT is related to longer falling pathway and more cloud droplets can be collected, consequently, thus heavier NSRR can be reached. However, the formation of ice and liquid cloud droplets, the growing mechanisms, and the collision efficiencies all can be modified by both dynamical, thermodynamical, and microphysical processes. Aerosols, either acting as CCN or IN also have potential capability to modulate the quantitative relationship between PTT and NSRR.

The PTT-NSRR relationships for stratiform, convective and warm rains in pristine (dotted line) and dusty (solid line) conditions, and for different levels of CAPE are shown in Fig. 9. For given NSRR, the PTT in pristine conditions is colder than those in the dusty conditions (i.e., the precipitation top is higher). For

given PTT, the precipitation rate NSRR under dusty conditions is stronger than that under pristine conditions. This confirmed that the changes in the microphysical processes (e.g., Figs. 7 and 8) induced by dust aerosol can lead to measurable changes in precipitation characteristics at the macroscopic scale.

CAPE is one of the most representative parameters of atmospheric dynamical conditions and can reflect the overall stability of the atmosphere, which was widely used to quantify the dynamical constraint on convection development in aerosol cloud interaction studies (e.g. Doswell and Rasmussen, 1994). We tested the impacts of CAPE on PTT-NSRR relationship in pristine samples to alleviate the tangling effects from aerosol.

All pristine stratiform and convective raining samples are divided into two groups with strong CAPE (i.e. over 700 J kg$^{-1}$ for stratiform rains and over 1100 J kg$^{-1}$ for convective rains) and weak CAPE (i.e. weaker than 350 J kg$^{-1}$ for stratiform rains and 700 J kg$^{-1}$ for convective rains) to check the impacts of dynamic conditions on the PTT-NSRR relationship. There are two criterions for the selection of CAPE thresholds. First, the differences between defined strong and weak CAPE groups should be great enough. Second, it is required that both groups have enough sample size. After the experiment, it was found that the threshold of weak (strong) CAPE was taken as 25% (55%) of the cumulative probability of CAPE for pristine raining samples as more appropriate. As shown in Fig. 9d-f, CAPE change the relationship between NSRR and PTT. For a given NSRR, PTTs under stronger CAPE (gray curves) is about 5-6 (2.0) degree colder than those under weaker CAPE (black curves) in both stratiform and convective rains (warm rains) under pristine conditions. The t-testing significances of difference exceeded the 99 % confidence level (Fig. S9). It indicates that strong dynamical conditions will favor raindrops reaching high altitudes with colder PTTs. Meanwhile, it was found the linear regression slopes of $K$ in Eq. (1) are similar between different CAPEs (Fig.9 d-f). It indicates the final rain rate reaching earth surface NSRR is proportional to the PTT with the same coefficient of $1/K$. In another words, the growth rates of rain drop along the falling path are similar under pristine environment.

Since the satellite observations showed PTT has a good linear relationship with NSRR (Fig. 9), the PTT can be expressed as a linear function of NSRR:

$$PTT = PTT_0 + K \times NSRR, \tag{1}$$

where $PTT$ is the precipitation top temperature (°C), $NSRR$ is the near surface rain rate (mm h$^{-1}$), $K$ is the linear regression slope, and $PTT_0$ is the intercept when NSRR equals to zero. Physically, $PTT_0$ represents

the PTT related to rain onset (when NSRR equals to zero), and $K$ represents the sensitivity of PTT to

NSRR. Previous investigations demonstrate that $K$ is relatively stable for different CAPES or aerosol

conditions (Dong et al., 2018; Li et al., 2011a), so that we mainly focus on the variations of $PTT_0$.

Cloud dynamics and aerosol-related microphysics combinedly affect the PTT-NSRR relationship. For

the difference of $PTT_0$ between two groups of samples with different CAPE and dust AOD, we separate

the difference of $PTT_0$ as a follows

$$\Delta PTT_0 = \frac{\partial PTT_0}{\partial CAPE}\Delta CAPE + \frac{\partial PTT_0}{\partial AOD}\Delta AOD, \tag{2}$$

where $\frac{\partial PTT_0}{\partial CAPE}$ represents the sensitivity of $PTT_0$ to CAPE, $\frac{\partial PTT_0}{\partial AOD}$ represents the sensitivity of $PTT_0$ to

aerosol.

Therefore, the term of

$$\frac{\partial PTT_0}{\partial AOD} = \frac{\Delta PTT_0 - \frac{\partial PTT_0}{\partial CAPE}\Delta CAPE}{\Delta AOD}. \tag{3}$$

To determine the sensitivity of $\frac{\partial PTT_0}{\partial CAPE}$, we randomly selected 70 % of the pristine precipitation samples

(to avoid potential contaminations from aerosol effect) to investigate the relationship between $PTT_0$ and

CAPE. All the samples were sorted into 6 bins with increasing values of CAPE, and the sample size for

each bin are general the same. In each bin, linear regression were conducted to determine K and $PTT_0$

following the Eq. (1). To determine the uncertainty of this estimation, we repeated the randomly selecting

processes 40 times. And all the results are shown in Fig. 10.

As we can see, the $PTT_0$ showed a very strong linear correlation with CAPE. The determining factor are

0.78, 0.86 and 0.31 for deep stratiform, convective and warm precipitation, respectively. With increasing

CAPE, cloud drops are easy to be elevated to higher altitude (low $PTT_0$ ) and precipitation embryos start

to form there.

Quantitatively, it was estimated that $PTT_0$ decreases by 0.41 °C per 100 J kg$^{-1}$ CAPE (the linear regression

slope in Fig. 10) for deep stratiform precipitation with variation of 12 % (standard deviation of the 40

times estimations of $\frac{\partial PTT_0}{\partial CAPE}$ divided by the mean $\frac{\partial PTT_0}{\partial CAPE}$). And $PTT_0$ decreases by 0.65 °C per 100 J kg$^{-1}$

CAPE for deep convective precipitation with variation of 15 %. For warm rains, $PTT_0$ decreases only by

0.066 °C per 100 J kg$^{-1}$ CAPE but with large variation of 38 % indicating this method didn't work good

for it.

Finally, substituting the estimated $\frac{\partial PTT_0}{\partial CAPE}$, the mean values of $\Delta PTT_0$, $\Delta AOD$, and $\Delta CAPE$ between dusty and pristine samples into the Eq. (3), the sensitivity of $PTT_0$ to aerosol optical depth $\frac{\partial PTT_0}{\partial AOD}$ was obtained. The $PTT_0$ increases by 4.19 °C per unit AOD for deep convective precipitation and by 0.35 °C per unit AOD for stratiform precipitation. Results for warm rains were not shown here due to its large uncertainties.

**4 Discussion and conclusion**

Mineral dust is a type of aerosol with the largest proportion of mass on land, and can act as both IN and CCN to affect clouds and precipitation. At present, the study of the indirect effects of dust aerosols on climate has shifted from qualitative to quantitative. It is expected that the effects of dust aerosols on clouds and precipitation can be accurately described in numerical models. A typical example is that Demott et al. (2010) directly wrote the number density of aerosols with an effective radius greater than 0.5 microns (mainly dust) into the IN parameterized formulas in cloud resolution mode. However, the effectiveness of such model parameterization is hard to be assessed with large-scale satellite observations. The physical characteristics of cloud or precipitation in real observations are affected by both aerosol indirect effects (if any) and atmospheric thermodynamical and dynamic effects. Unfortunately, to isolate those two types of effects is the most difficult part of observational study on the cloud aerosol interaction. In this study, we selected southeast China, which is far from the original source of dust for study area. Here, heavy dust conditions are often accompanied by strong northwesterly winds, strong wind shear, cold air temperature, and weak CAPE. Such relatively fixed weather conditions facilitate the isolation of aerosol indirect effects from dynamic effects. In order to study the IN effect of dust, we selected the vertical profile of precipitation observed by the spaceborne rain radar measurements as the basic data. By coupling multi-source satellite data and reanalysis data, dusty precipitation samples and pristine precipitation samples in June, July and August (JJA) during 2000 to 2013 were separated for comparative analysis.

First, it was found that there was a difference in the averaged vertical profile between dusty and pristine samples. For the same PTT, the dusty precipitation rate was weaker than that of pristine precipitation in the upper layer; and in the middle layer, it was significantly larger than the pristine precipitation, and the precipitation rate near surface was also larger under the conditions of dust. By quantifying the rate of

precipitation growth, it is found that in the upper layers, the growth of dusty precipitation is slower. But in the middle layer, the growth rate of dusty precipitation is remarkably faster than that of pristine precipitation. Qualitatively, these phenomena confirmed that mineral dust can enhance the heterogeneous freezing process and lead to observable changes in precipitation rate.

In this study area, heavy dusty conditions is always coupled with strong wind shear, which favors evaporation process in the lower layers (temperatures higher than 2 °C) of stratiform precipitation, corresponding to a stronger cooling. Stronger wind shear is also unfavorable to the warm rain process due to suppression of cloud droplet coalescence. Dust aerosols may delay the onset of weak precipitation in the lower layers, thus more water droplets are lifted to the middle layers (temperature -5 °C to 2 °C) where dust aerosols can act as additional ice nuclei to enhance heterogeneous freezing. Consequently, at middle layers, precipitation particles grow rapidly through aggregation and riming, releasing large amounts of additional latent heat.

As a macroscopic manifestation of the above microphysical processes, we found that for a certain surface precipitation rate, the PTT of dusty precipitation is higher than that of pristine precipitation. This is a combination of specific weather conditions and potential aerosol indirect effects on dust days. Through data analysis, we found that for every 100 J kg$^{-1}$ decrease in CAPE, the onset PTT of convective precipitation and stratiform precipitation will increase by 0.65 and 0.41 degrees Celsius respectively. This is the first reason for the higher PTT on dusty days. In addition, for every unit increase in the aerosol optical depth of dust aerosols in the atmosphere, the PTT of convective precipitation and stratiform precipitation will increase by 4.19 and 0.35 degrees Celsius respectively. This is the second reason for the higher temperature of the rain top of precipitation on dusty days.

Based on our knowledge, this is the first time to separate the contribution of aerosol to satellite observed PTT from real satellite observations. The results can be used to evaluate cloud resolving model simulations and to assess the performance of model parametrizations related to dust aerosol IN effect.

Although the heavy dusty days in southeastern China is not frequent, the generally accompanied synoptical pattern provides us ideal testbed to isolate the dust aerosol effects from dynamic effects on precipitation. And the associated mechanism and effetcs should be also valid in other region, where the dynamic effects may not be isolated easily. And the method of isolating the influence of dynamical and aerosol conditions on cloud precipitation can be applied to other regions.

It should be noticed there are several uncertainties in this study. There are uncertainties in the MODIS retrieval of aerosols over land (Chu et al., 2002), and the uncertainty in the FMF retrieval is about ±0.2 (Tanre et al., 1996; Tanre et al., 1997). There is still a lack of long-term, large-scale dust observation product to solve this problem precisely. Instead, multiple studies were conducted based on MODIS retrieved FMF information. For example, Kaufman et al. (2002, 2005) and Gao et al. (2001) have utilized the FMF-derived CMAOD as representation of dust to study the transport and deposition of dust and its impact on the climate system. Min et al. (2009) and Li et al. (2010) applied MODIS derived coarse mode AOD to classify dust aerosols over Atlantic Ocean to study their impacts on cloud and precipitation profiles. The Cloud-Aerosol Lidar and Infrared Pathfinder Satellite Observations (CALIPSO) Level 2 lidar vertical feature mask (VFM) data product uses the particle depolarization ratio to determine the dust. However, CALIPSO only has nadir observations, and the data obtained from narrow orbits are very limited. Therefore, we did not use the CALIPSO data as the basis for judging dust in this study. However, it can be used as a supporting evidence for the adoption of CMAOD by MODIS to determine dust. For example, CMAOD shows a typical dusty precipitation day on June 25, 2011 and July 9, 2011, and CALIPSO's VFM product likewise shows that the aerosols on that day were indeed predominantly dust (Fig. S10). We performed a sensitivity test assuming that there is a random error of up to ±20% in CMAOD and that the PTT-NSRR relationship for the new data (Fig. S11) and the original data (Fig. 10) remain unchanged. That is, there is some error in CMAOD, but it does not subvert the conclusions of this study. In addition, in this study, we have not considered the aerosol humidification effect in the presence of precipitation, which may increase the retrieving error of FMF in MODIS aerosol product. Firstly, the MODIS algorithm filters out pixels within 1 km of detectable clouds, where the effect of aerosol humidification will be the greatest (Martins et al., 2002). And this algorithm significantly reduces the effect of relative humidity on aerosol optical depth retrievals (Remer et al., 2005). Secondly, Altaratz et al. (2013) performed radiative transfer calculations using 12 years of June-August radiosonde measurements and found that at continental stations, the AOD increased by 4% and 5% for the 1 km and 2 km layers, for aerosol hygroscopicity = 0.3, respectively, and by 5% and 4% for aerosol hygroscopicity = 0.7. That is, the effect of changes in relative humidity on AOD is limited. In our study, assuming a 5% hygroscopic growth of AOD, the relative increase of $\frac{\partial PTT_0}{\partial AOD}$ for stratiform (convective) precipitation is 2.8% (3.3%). Such effect will not significantly change our conclusion. In addition, the relationship

between NSRR and PTT is influenced by multiple dynamic factors. Sensitivity tests of $PTT_0$ to updraft velocity (W), water vapor (RH) and wind shear were conducted using the same method for CAPE

(Figs. S12, S13 and S14). The relationship of $PTT_0$ to them at 750 and 500 hPa are not as stable and significant as that to CAPE. This is because the PTH varied from case to case and is sensitive to multiple factors at varied altitudes. CAPE as a measure of the convective instability energy has the best representativeness of dynamic effects on precipitation vertical structure. Therefore, in this study, we mainly focused on CAPE.

**Data availability.** Precipitation data are obtained from the Tropical Rainfall Measuring Mission (TRMM) satellite (https://gpm.nasa.gov/missions/trmm). The standard TRMM 2A25-base latent heat products derived from Convective and Stratiform Heating algorithm (CSH) and Spectrum Latent Heating algorithm (SLH) are available at https://search.earthdata.nasa.gov/search?q=TRMM. The latent heat

products derived from vertical profile heating algorithms (VPH) are available from Hongxia Zhu (zhx227@mail.ustc.edu.cn). Aerosol optical thickness data are obtained from Moderate Resolution Imaging Spectroradiometer (MODIS) on Terra satellite (https://modis.gsfc.nasa.gov/). The data for the back trajectory analysis are obtained from Hybrid Single-Particle Lagrangian Integrated Trajectory (HYSPLIT) model (https://www.ready.noaa.gov/hypub-bin/trajtype.pl?runtype=archive). The Cloud-

Aerosol Lidar and Infrared Pathfinder Satellite Observations (CALIPSO) Level 2 lidar vertical feature mask (VFM) data products are available at https://subset.larc.nasa.gov/calipso/. Hourly meteorological data are obtained from European Centre for Medium-Range Weather Forecasts ERA5 reanalysis (https://www.ecmwf.int/).

**Author Contribution.** R. Li designed the experiments and H.X. Zhu and S.P. Yang carried them out. S.P. Yang and C. Huang conducted the latent heating retrieving. H.X. Zhu and R. Li prepared the manuscript with contributions from all co-authors.

**Competing interests.** The authors declare that they have no conflict of interest.

**Acknowledgements.** This work was supported by the National Natural Science Foundation of China (Grant No. 41830104, 41661144007, 41675022), National Key Research and Development Program of China (Grant No. 2021YFC3000300), and the Jiangsu Provincial 2011 Program (Collaborative Innovation Center of Climate Change).

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

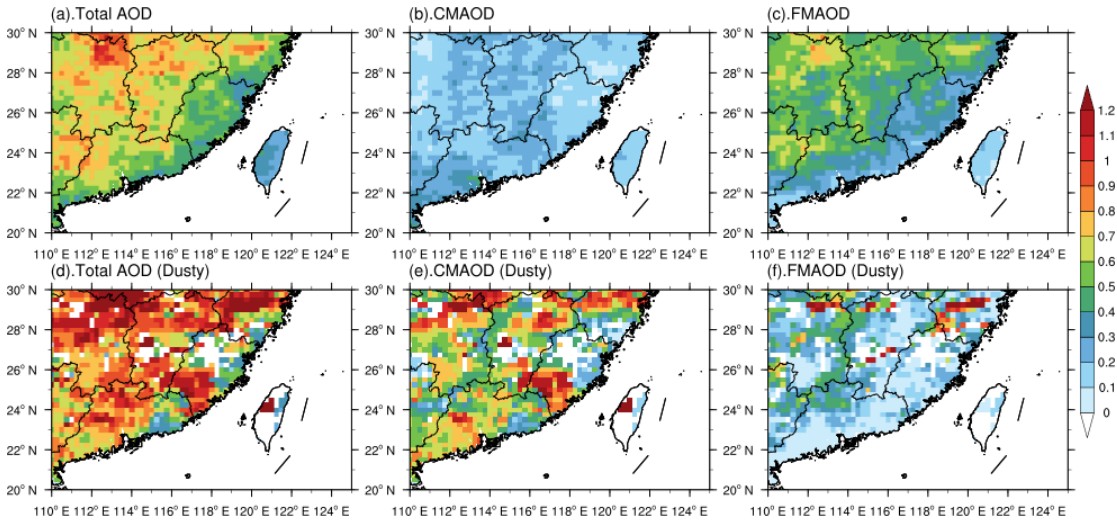

**Figure 1: Total aerosol optical depth (a, d), coarse mode aerosol optical depth (CMAOD, b, e), and fine mode aerosol optical depth (FMAOD, c, f) under all days (the first row), and dusty days (the second row) in June, July and August (JJA) during 2000 to 2013 retrieved from Terra MODIS.**


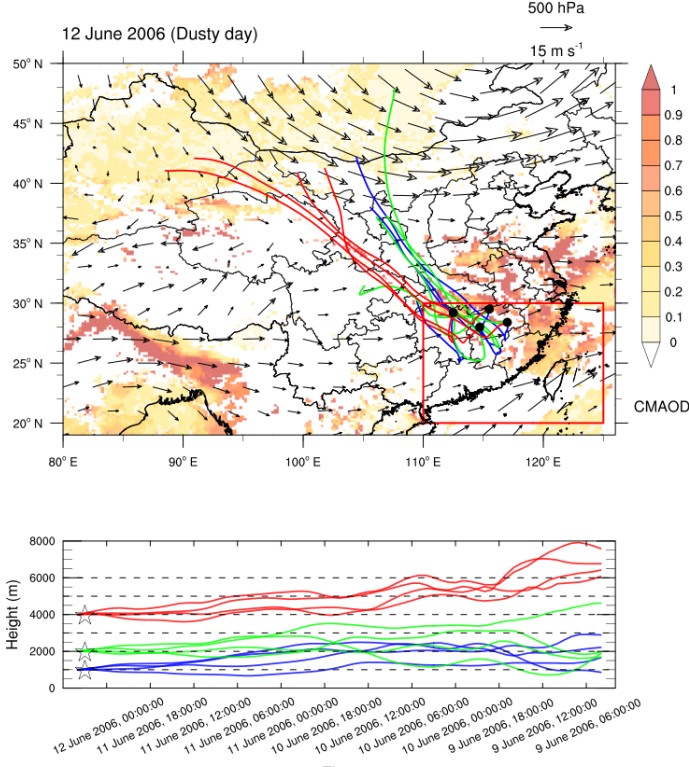

**Figure 2: Horizontal distribution of coarse mode aerosol optical depth derived from Terra MODIS, wind fields at 500 hPa, and 72-hour back trajectories from the HYSPLIT model on 12 June 2006. Where the red box indicates the study area, the geolocation of four starting points are at 29.5° N, 115.5° E; 28° N, 114.7° E; 28.4° N, 117° E; and 29.2° N, 112.5° E, with altitudes of 1000 m (blue line), 2000 m (green line), and 4000 m (red line), extrapolated from 12 June 2006 at 04:00 UTC.**


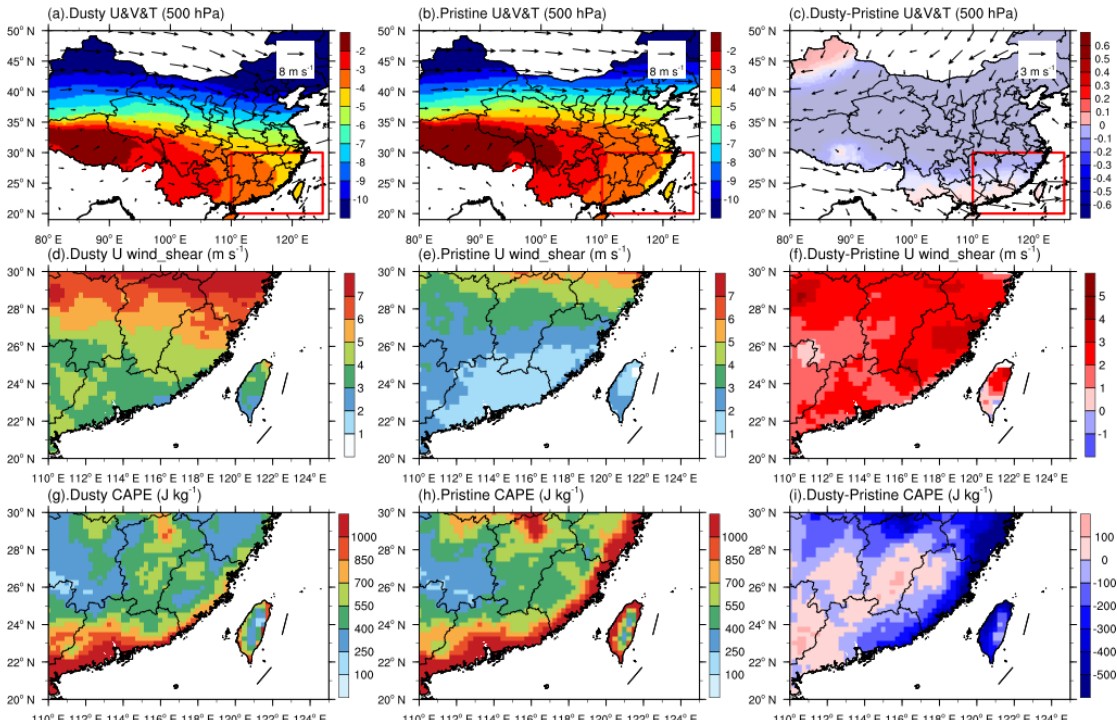

Figure 3: The fields of wind and temperature at 500 hPa (upper row), U wind shear (middle row), CAPE (bottom row) averaged from selected 46 dusty days (left column) and 92 pristine days (middle column) in JJA during 2000-2013 based on ERA5 reanalysis data at horizontal resolution of 0.25°×0.25°, and the associated differences between them (dusty minus pristine, right column).

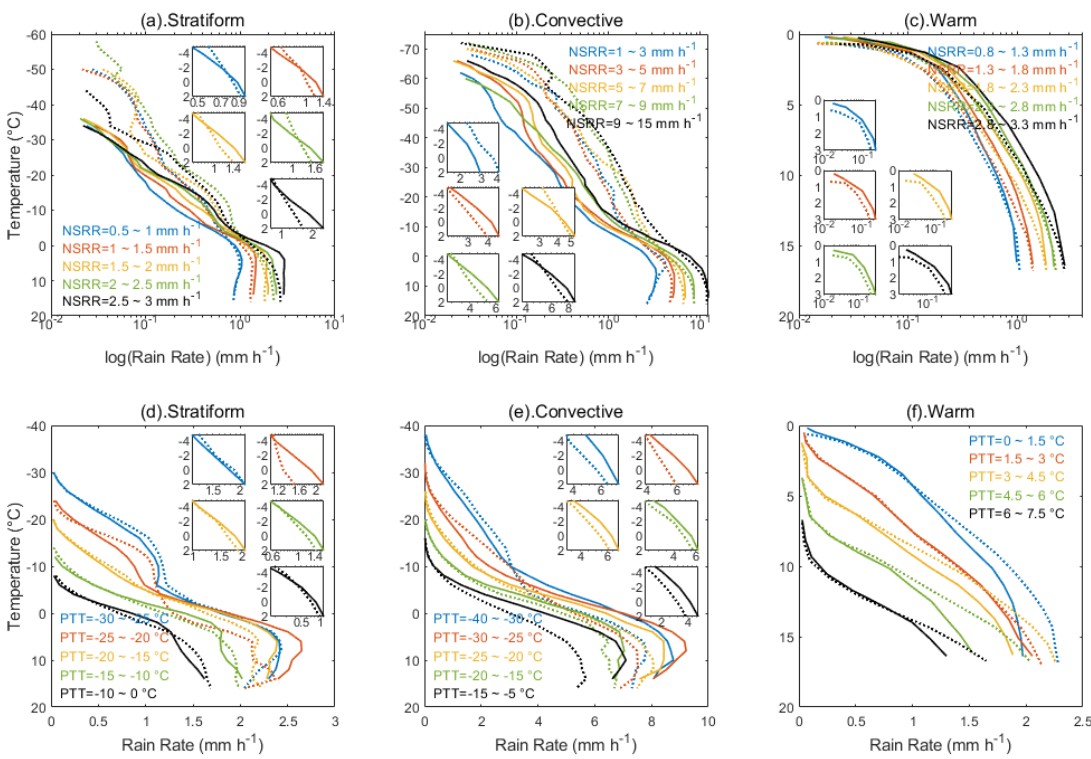


**Figure 4: Differences in vertical profiles of stratiform (a,d), convective (b,e) and warm (c,f) precipitation for pristine (dashed line) and dusty (solid line) conditions for given NSRR (the first row) and PTT (the second row). Different color stands for different NSRR and PTT. Each subpanel focuses on the rain rate in the mixed layer (temperatures between -5 °C to 2 °C).**


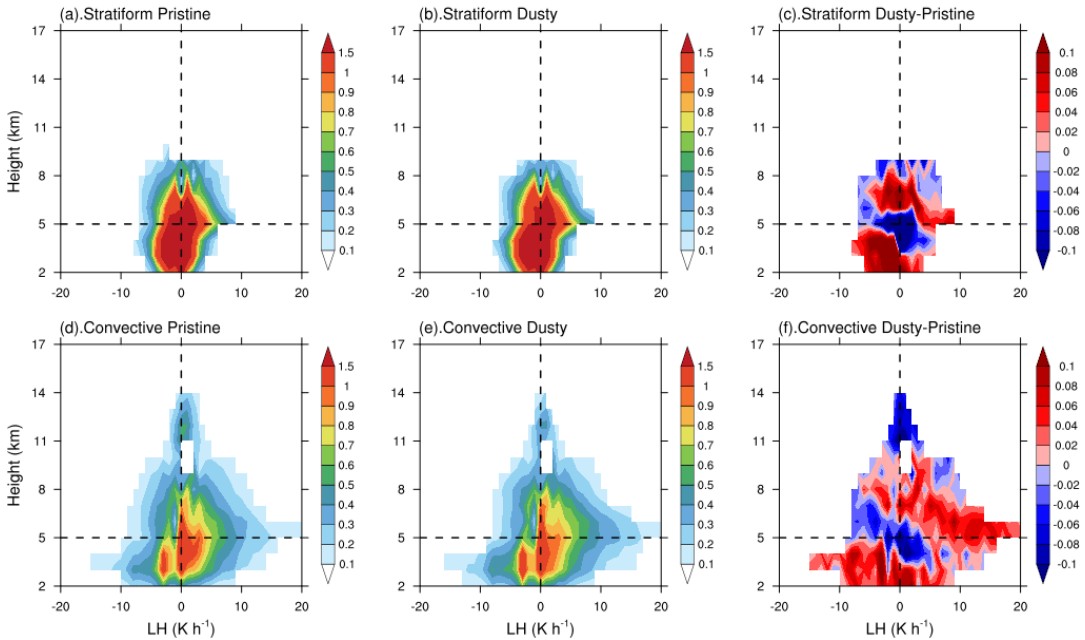

**Figure 5: Contoured frequency by altitude diagrams (CFADs) of LH (retrieval from VPH) in pristine (the first column) conditions, dusty (the second column) conditions and the differences between them (the third column) for stratiform (the first row) and convective (the second row) rains.**


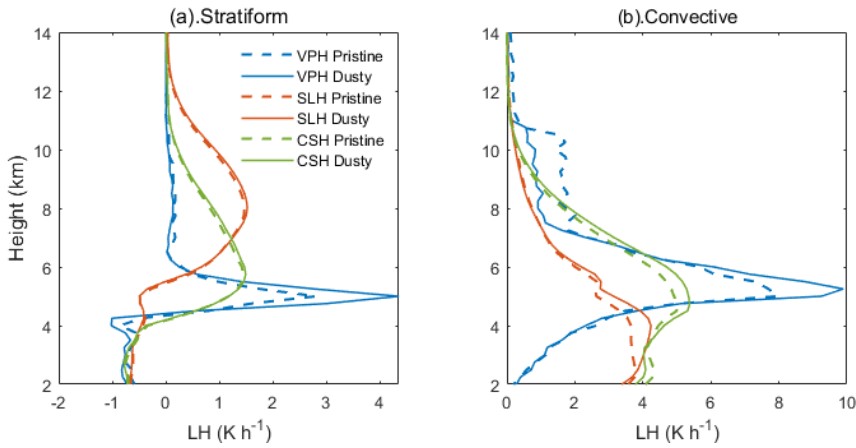

**Figure 6: The mean latent heating (LH) profiles retrieved from VPH (blue), SLH (red) and CSH (green) for (a) stratiform precipitation and (b) convective precipitation in pristine (dashed) and dusty (solid) conditions.**

Stratiform

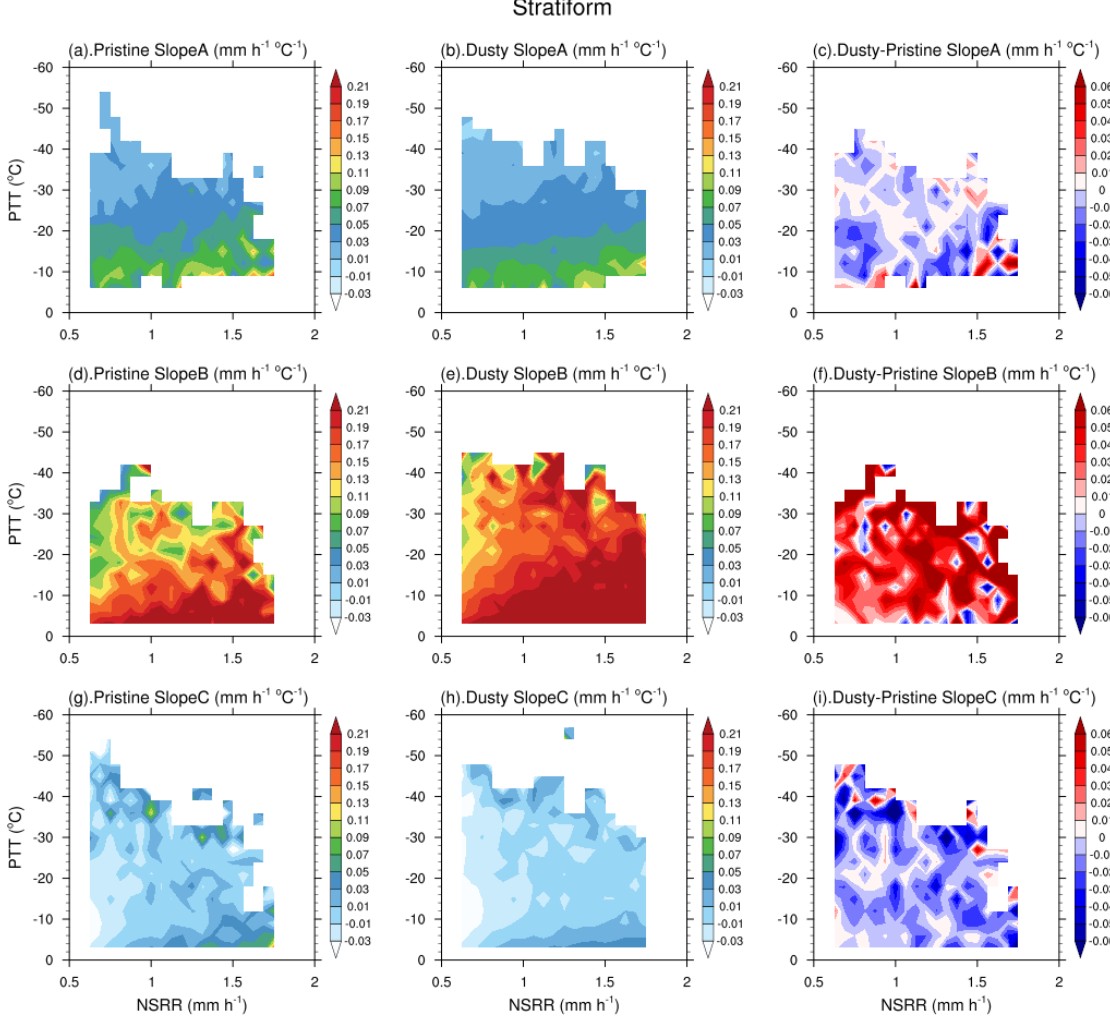


**Figure 7: The mean SlopeA (the first row), SlopeB (the second row), and SlopeC (the third row) for stratiform precipitation as functions of near surface rain rate (NSRR) and precipitation top temperature (PTT) in pristine (the left column) conditions, dusty (the middle column) conditions and the differences between them(dusty minus pristine, the right column).**


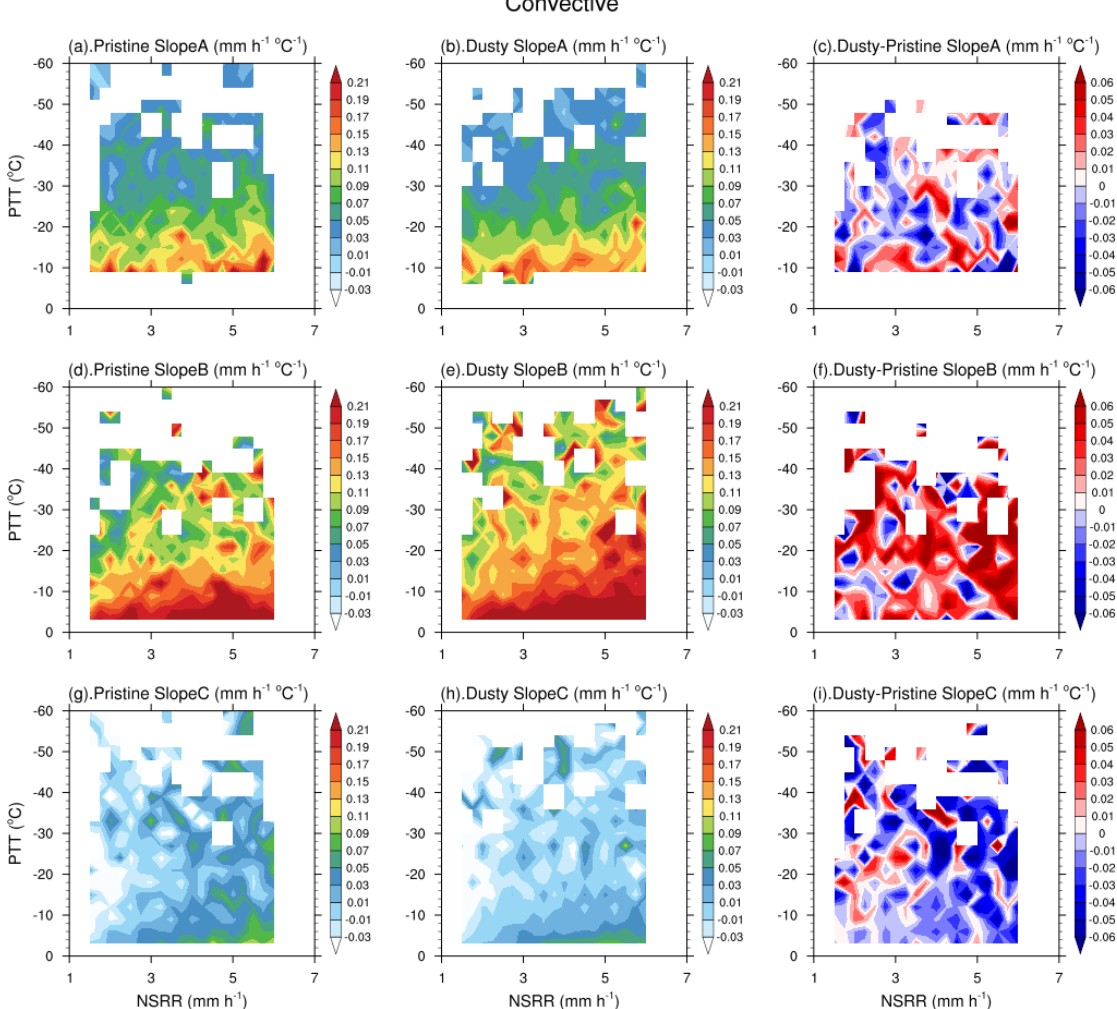

**Figure 8: As same as Figure 7, but for deep convective precipitation.**

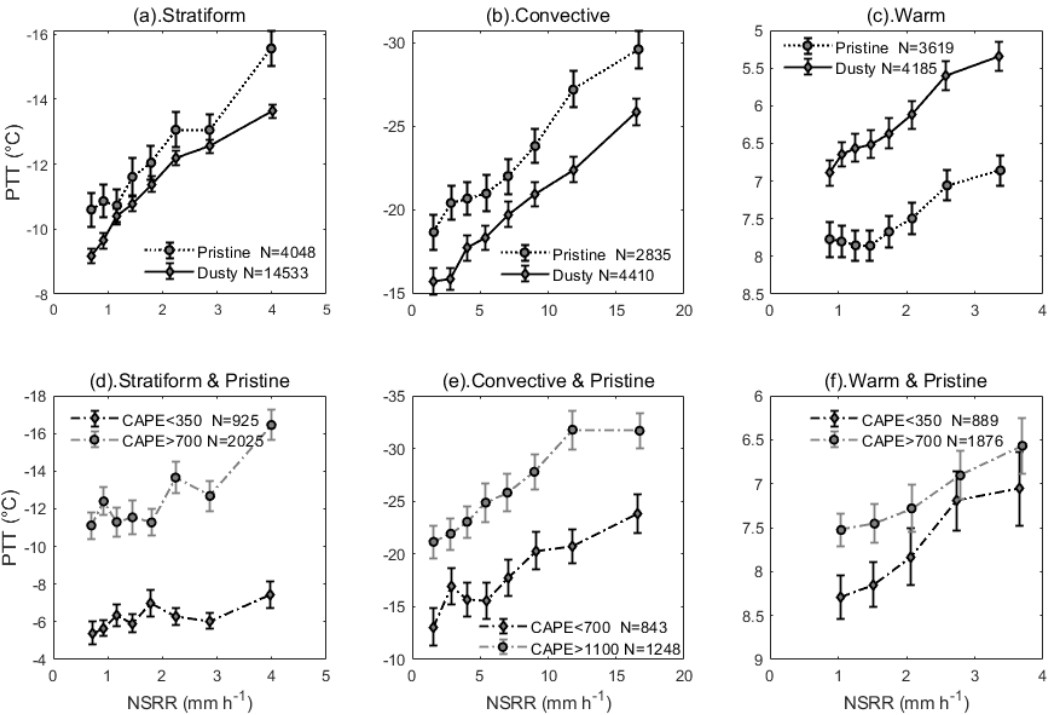

Figure 9: The precipitation top temperature (PTT) against near surface rain rate (NSRR) for stratiform (a, d), convective (b, e) and warm (c,f) precipitation under pristine (dotted curves) and dusty (solid curves) conditions (the first row) and in different CAPE (black line: weak CAPE, gray line: strong CAPE) conditions under pristine conditions (the second row).

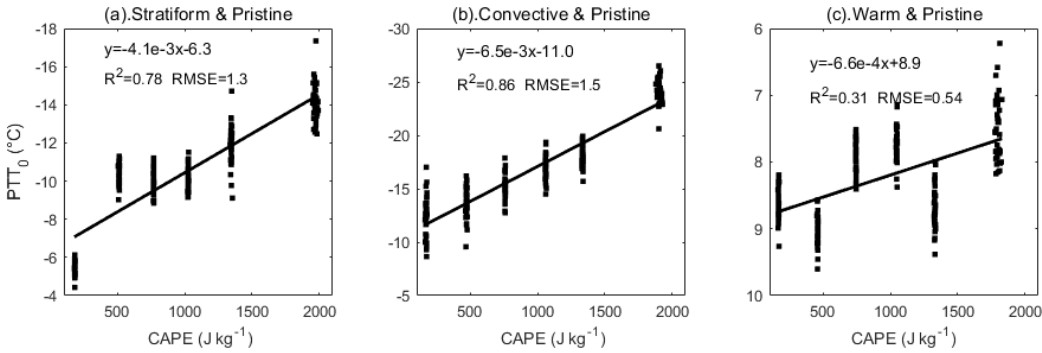


**Figure 10: The variation of $PTT_0$ with CAPE under pristine conditions for (a) deep stratiform precipitation; (b) deep convective precipitation and (c) warm rains. The results are derived from randomly selected 70% precipitation samples from total.**