# Peer review of "The impacts of dust aerosol and convective available potential energy on precipitation vertical structure in eastern China as seen from multiple source observations"

_Atmospheric Chemistry and Physics, 2022_

## Author Response (AR1)

**Answers to anonymous referee #1**

We would like to thank the reviewer for his/her time and effort reviewing our study. We have found the comments to be constructive and helpful.

In this reply, the comments from the reviewer are in black, and our answers are in red. The new text and lines of the revised document where the adjusted text can be found are also in red. In the revised document, all new text is marked in blue, and deleted text is crossed out in red.

**Major comments:**

1. Abstract: This work mainly investigated the dust aerosol impact on precipitation vertical structures using multi-source data, and gave statistical results. Nevertheless, I did not see the specific study area and time period.
   A: We have added the specific study area and time period to the abstract.

   "Abstract. The potential impacts of dust aerosol and atmospheric convective available potential energy (CAPE) on the vertical development of precipitating clouds in southeastern China (110° E-125° E; 20° N-30° N) in June, July, and August during 2000 to 2013 were studied using multiple-source observations."

2. The titile of Section 2 of "Study area and data" needs to be rephrased, since most of the paragraphs focus on the methodology. As such, this section can be restructured. For instance, I do not understand what is the logic and purpose of the references such as "Teller and Levin (2006)" and Yin and Chen (2007), both of which are simply listed as seprated arguments and not tightly linked to the data or methodologies used in this study. In my opinion, these descriptions are more like related to the research status and can be moved to the introdution part.
   A: We agree with this comment. Since the two references are not tightly linked to our study, we decided to delete them in the revision. And we changed the title of Section 2 to be "Data and Method".

3. The English writing of this manuscript needs thorough improvement, and a complete polishing from the abstract to conclusion part is necessary with the help of a native English speaker or a more experienced researcher.
   A: We agree with the reviewer, and we will have a native English speaker to polish the revised manuscript.

4. L204-206: In the presence of dust episode occurring in eastern China, a combination of high wind shear, low cape was observed, the author argued that "Such condition doesn't favor the vertical development of convection." Are there any literatures supporting this argument? In my view of point, the precipitation accompanied with dust episode is largely under the influence of large-scale circulation. This synoptic forcing favors the lifting of air mass and convection initiation. Besides, not every

precipitation event was characterized by high CAPE, low wind shear.

A: We agree with the reviewer that precipitation in dusty days in southern China is largely influenced by large-scale circulation. And this synoptic forcing favors the lifting of air mass and convection initiation comparing to that in non-raining days.

The description "high wind shear and low CAPE, such condition doesn't favor the vertical development of convection" is an inaccurate expression on our part. What we want to express is that the background field of precipitation in dusty days has a lower CAPE and stronger wind shear compared to the pristine days. Low CAPE (Rosenfeld et a1., 2008) and strong wind shear (Fan et al., 2009, 2013) were found to be detrimental to the development of convection in other study areas.

Indeed, only from long-term statistics, pristine precipitation events are featured by relatively high CAPE and low wind shear conditions. There are considerable variations on the relationship between PTT and CAPE as shown in Section 3.

We clarified all above concerns in the revision as:

"In both dusty and pristine precipitation days, the synoptic forcing conditions favor the lifting of air mass and convection initiation comparing to that in non-precipitating days. Statistically, pristine precipitation events are featured by relatively higher CAPE and lower wind shear conditions, which may enhance the vertical development of the precipitating clouds."

**Minor comments:**

1. "eastern China" in the title of this manuscript can be revised to "southeastern China"
   A: Yes, and we have changed it in the manuscript.

2. L14: "the study area" is suggested to be replaced with a specified area (e.g., southeast China?)
   A:Yes, and we have changed it in the manuscript.

3. L15: "contained"-> "containing"
   A: Yes, and we have changed it in the manuscript.

4. L34: ", they can" -> ", which can"
   A: Yes, and we have changed it in the manuscript.

5. L35: "to directly affect" -> "thereby directly affecting"
   A: Yes, and we have changed it in the manuscript.

6. L38: "warmer temperature" is not appropriate and can be revised to "higher temperature"

A: Yes, and we have changed it in the manuscript.

7.  L39: "server" -> "serve"
    A: Yes, and we have changed it in the manuscript.

8.  L40: "moderate" can be revised to "mediate" or "modulate"
    A: Yes, and we have changed it in the manuscript.

9.  L46: "Studies" -> "Previous studies"
    A: Yes, and we have changed it in the manuscript.

10. L78:"significantly was" -> "was significantly"
    A: Yes, and we have changed it in the manuscript.

11. L79: ". Such as" -> ", including"
    A: Yes, and we have changed it in the manuscript.

12. L83: I would suggest adding more recent references on the dependence of aerosol effect on precipitation on "the altitudes of the aerosol layer", and the authors can refer to Lee et al. ACPD 2022 (https://doi.org/10.5194/acp-2022-385) and the references therein.
    A: Thanks, and we have added the reference to the manuscript.

    "Aerosol-cloud-precipitation interactions (ACIs) also largely depend on meteorology conditions including wind shear (Fan et al., 2009, 2013), atmospheric stability (Huang et al., 2014), relative humidity (Li et al., 2019b), and the altitudes of the aerosol layer (Yin et al., 2012, Lee et al., 2022)."

13. L104: "In some studies," in which studies? The authors can add references here to support this statement.
    A: Since this statement is not tightly related to this study, we decided to remove it from the manuscript.

14. L132: grammar errors in ", they are treated".
    A: Yes, and we have changed it in the manuscript

15. L137: The acronym for "precipitation top temperature" has been given in introduction part and in this place and the following section, it is supposed to appear as "PTT".
    A: Yes, and we have changed it in the manuscript.

16. L173: the references to ERA-5 reanalysis are lacking.
    A: We have added the references in the manuscript. And we have clarified this point as:

"The atmospheric thermodynamic conditions under pristine or dusty environments were derived from hourly ERA5 reanalysis data at horizontal resolution of 0.25°×0.25° (Hersbach et al., 2020)."

17. L178: "during recent two decades" -> "during recent decades"
A: Yes, and we have changed it in the manuscript.

18. L179-180: it is a too long sentence in "by anthropogenic emission related fine mode aerosols with small fraction of coarse mode aerosol", and full of redundant words. I would suggest rewriting
A: Yes, and we have changed it in the manuscript. We have clarified this point in the revision as:

"In summer, the area generally was dominated by anthropogenic emissions of fine mode aerosols."

19. L181: "in which" or "when" can be added before "heavy dust aerosol"
A: Yes, and we have changed it in the manuscript.

20. L182: "are" -> "were", and "that" is missing before "were defined"
A: Yes, and we have changed it in the manuscript.

21. L183: "excessed" -> "exceeded"
A: Yes, and we have changed it in the manuscript.

22. L188-190: I would suggest clarifying whether the selected date of 12 June 2006 was also a rainy day in the southeast China.
A: Yes, it is a raining day and this is clarified in the revision as:

"For example, on 12 June 2006 a typical dusty precipitation day, about half of the study area was covered by heavy dust (Fig. 2) with satellite observed CMAOD up to 1."

23. L199: "CAPE" is not an atmospheric dynamic variable
A: We clarified the associated statement as:

"Finally, as an overall measure of regional mean atmospheric insatiability, regional CAPE was 600 J kg$^{-1}$ in dust conditions and 743 J kg$^{-1}$ in pristine conditions."

24. L200: please clarify what is "strong coupling" between dusty condition and meteorology condition? It seems to me this term is contradictory with the following weak correlation observed between dust AOD and meteorology.
A: Yes, the statement is not precise enough, so we refined it as:

"In summary, in southeastern China, heavy dusty condition is generally accompanied by certain synoptic pattern dominated by strong north wind."

25. L225: "start"-> "starting"
    A: Yes, and we have changed it in the manuscript.

26. L288 and L321: the full name for precipitation top height was actually given in L66, and thus should be avoided here.
    A: Yes, and we have changed it in the manuscript.

27. L362: Except for the "atmospheric thermodynamical effects", the atmospheric dynamic impact can not be ignored.
    A: We have changed it in the manuscript, and now it was clarified as:

    "The physical characteristics of cloud or precipitation in real observations are affected by both aerosol indirect effects (if any), atmospheric thermodynamical and dynamic effects."

28. Figure 3: I would suggest adding the time period for which the meteorological fields are derived. Also the data sources are suggested to be added in this figure caption.
    A: Thanks for this valuable suggestion. We modified the caption of Figure 3 as:

    "Figure 3: The fields of wind and temperature at 500 hPa (upper row), U wind shear (middle row), CAPE (bottom row) averaged from selected 46 dusty days (left column) and 92 pristine days (middle column) in JJA during 2000-2013 based on ERA5 reanalysis data at horizontal resolution of 0.25°×0.25°, and the associated differences between them (dusty minus pristine, right column)."

29. Figure 11: it would be beneficial to give more descriptions in the figure caption on "70%" at the top of each panel in this figure. What does it mean, or how is it defined.
    A: To avoid confusion, we have deleted the "70%" from the Figure, and we have modified the Figure caption as:

    "Figure 11: The variation of $PTT_0$ with CAPE under pristine condition for (a) deep stratiform precipitation; (b) deep convective precipitation and (c) warm rains. The results are derived from randomly selected 70% precipitation samples from total."

[Figure]

[Figure]

[Figure]

An elaboration has been added about the main challenges facing the study of aerosol-cloud-precipitation interactions and how scientists are trying to solve the problem.
- "A great challenge in observational study on the indirect effects of aerosols is to distinguish the isolated contributions of weather conditions (dynamic conditions) and aerosol microphysical effects to the observed macro-micro features of clouds and precipitation (Stevens and Feingold 2009; Tao et al., 2012; Rosenfeld et a1., 2014; Li et al., 2017). This is especially true for mesoscale convective systems (MCSs) that are heavily affected by large-scale atmospheric circulation. Some studies have adopted this ideals to constrain the variations of dynamical factors, cloud type, stages of cloud precipitation development and etc., and then to analyze the influence of aerosols (Rosenfeld et al.,2008; Fan et al.,2013, 2018; Li et al., 2011b; Min et al.,2009; Li and Min,2010; Gibbons et al., 2018). For example, Fan et al. (2013) found that the thermodynamic effect of aerosols (freezing of cloud water to release additional LH) contributes up to 27 % to the increase in cloud cover during the growth stage of deep convective clouds in summer, while the

microphysical effect of aerosols (freezing of large amounts of cloud droplets to produce more and smaller ice particles) increases cloud cover and cloud top height during the mature and dissipation stages."

We have also added explanations why this study is novel.
- "And we attempt to isolate the impacts from meteorology conditions and aerosol conditions on the vertical structure of precipitation and LH by analyzing multiple satellite observation with new mathematic treatment."

2. section 3.2 is important and actually contains quite a lot interesting findings. But only the simple descriptions were presented without giving any discussion, implication or even comparison with previous studies. After reading this section, I don't really get scientifically useful information. It's more like a technical report.
A: Yes, we have added more discussions, comparisons with results in other studies, and some hypotheses into this section. The revised statements are as shown here:
- "Although followed by a layer with slower growing, the final NSRR for given PTT under dusty condition (solid curve) still is heavier than that of pristine rains (dotted curve). Such effect is weak for stratiform rains particularly those with relatively warm PTTs (e.g. light blue and green curves in Fig. 4d). This is because the proposed dust's IN effect generally works for ice-phase microphysical process. For those stratiform rains start from warm PTTs, there is no sufficient water content and the temperatures are too warm for the heterogeneous freezing to take place."

- "This indicates a possible suppression by dusty conditions for warm rain growth. During the long-range transportation of dust from north to southeastern China, very likely the dust particles were coated by soluble aerosols and become active CCN (Li et al., 2010) in the warm rains. For given condensed liquid water content, this additional CCN leads to smaller cloud effective radius thus decreases the coalescence efficiency which is the main mechanism for warm rain growth (Rosenfeld et al., 2008; Min et al., 2009; Yin and Chen, 2007; Li et al., 2010)."

- "Validation of satellite retrieved LH is still a very challenging task (Tao et al., 2022) because there is no directly measured ground-truth of LH available. Intercomparison among different LH products is one of the useful indirect means to evaluate their accuracy. Based on Li et al., (2019a), VPH product showed reasonable structure of LH in Tibetan Plateau with similarities and dissimilarities comparing to CSH and SLH. In this study, the VPH product was chosen because it is directly related to the variations of precipitation rate at each altitude, while CSH and SLH retrievals use constrains of precipitation rate at surface, precipitation top height, precipitation type, etc. It should be emphasized, the LH-related results did not receive rigorous validation in this

study area, thus should be treated with cautions."

**Specific comments:**

1. Line 16: How did author define the 'pristine days'? It is incorrectly used if the authors only meant days with low dust concentrations, because other aerosol can dominate especially over east China.
   A: We clarified this point in the revision:
   "If the mean total AOD is less than 0.2, the day was defined as pristine day."

2. In lines 49-57, the author showed the findings of dust aerosol weakening convection precipitation but immediately in lines 59-65 the opposite was listed. I would expect at least an explanation / mention here.
   Line 65: Yes, we added associated explanations in the revision as:

   "Observational and model simulation studies have shown different results for aerosol effects on deep convection, suggesting that aerosols may either invigorate or inhibit precipitation, depending on the type and concentration of aerosols and environmental conditions (Jiang et al., 2018; Khain 2009; Fan et al., 2009, 2013; Rosenfeld et al., 2008, 2014)."

3. Lines 133-136: this is a repetition of lines 130-132.
   A: Thanks for reminding us, and lines 130-132 have been deleted.

4. MODIS-retrieved aerosol size parameters have little quantitative skill over land (e.g., https://doi.org/10.5194/amt-4- 201-2011). Thus, derivation of CMAOD from FMF is not a good try.
   A: Yes, we agree. We clarified this point in the discussion part with three comments. Firstly, there has been a lot of literature using CMAOD to represent the AOD of dust. Secondly, we verified that the CMAOD of MODIS is dust using CALIPSO's aerosol and cloud vertical and horizontal distribution products (vertical feature mask product). Finally, we performed sensitivity tests to randomize the CMAOD to produce errors within ±20%, and the results showed that changes in the CMAOD do not have a subversive effect on our conclusions.

   We clarified this point in the discussion part in the revision as:
   ● "There are uncertainties in the MODIS retrieval of aerosols over land (Chu et al., 2002), and the uncertainty in the FMF retrieval is about ±0.2 (Tanre et al., 1996; Tanre et al., 1997). There is still a lack of long-term, large-scale dust observation product to solve this problem precisely. Instead, multiple studies were conducted based on MODIS retrieved FMF information. For example, Kaufman et al. (2002, 2005) and Gao et al. (2001) have utilized the FMF-derived CMAOD as representation of dust to study the transport and deposition of dust and its impact on the climate system. Min et al. (2009) and Li et al.

(2010) applied MODIS derived coarse mode AOD to classify dust aerosols over Atlantic Ocean to study their impacts on cloud and precipitation profiles."

● "The Cloud-Aerosol Lidar and Infrared Pathfinder Satellite Observations (CALIPSO) Level 2 lidar vertical feature mask (VFM) data product uses the particle depolarization ratio to determine the dust. However, CALIPSO only has nadir observations, and the data obtained from narrow orbits are very limited. Therefore, we did not use the CALIPSO data as the basis for judging dust in this study. However, it can be used as a supporting evidence for the adoption of CMAOD by MODIS to determine dust. For example, CMAOD shows a typical dusty precipitation day on June 25, 2011 and July 9, 2011, and CALIPSO's VFM product likewise shows that the aerosols on that day were indeed predominantly dust (Fig. S7)."

[Figure]

Figure S7: On June 25, 2011 and July 9, 2011, the vertical and horizontal distribution of cloud and aerosol layers observed by the Cloud-Aerosol Lidar and Infrared Pathfinder Satellite Observations (CALIPSO) lidar vertical feature mask (VFM) data product. Where the blue line in indicates the CALIPSO footprint.

● "We performed a sensitivity test assuming that there is a random error of up to ±20% in CMAOD and that the PTT-NSRR relationship for the new data (Fig. S8) and the original data (Fig. 10) remain unchanged. That is, there is some error in CMAOD, but it does not subvert the conclusions of this study."

[Figure]

Figure S8: The precipitation top temperature (PTT) against near surface rain rate (NSRR) for new stratiform (a) , convective (b) and warm (c) precipitation samples under pristine (dotted curves) and dusty (solid curves) conditions (the first row). For a given NSRR, t test significance for differences in PTT between stratiform (d), convective (e) and warm (f) precipitation in pristine and dusty conditions (the second row), red (black) line indicates the 95 % (99 %) confidence level at 100 degrees of freedom.

5. In addition, how did the author consider the aerosol humidification effect in the presence of precipitation.

A: Yes, we agree with the reviewer. And we admitted that aerosol humidification effect is important. In this study, the effect may increase the retrieving error of FMF in MODIS aerosol product, however, we have not considered such effect. Firstly, the MODIS algorithm filters out pixels within 1 km of detectable clouds, where the effect of aerosol humidification will be the greatest (Martins et al., 2002). And this algorithm significantly reduces the effect of relative humidity on aerosol optical depth retrievals (Remer et al., 2005). Secondly, the relationship between aerosol hygroscopic growth and the surrounding relative humidity values can be described by a single parameter representation, namely the kappa parameterization (Petters and Kreidenweis, 2007):

$$g(\kappa, \mathrm{RH}) = \left(1 + \kappa \cdot \frac{RH}{100 - RH}\right)^{1/3},$$

where g is the hygroscopic growth factor, $\kappa$ is the aerosol hygroscopicity (atmospheric particulate matter is typically characterized by $0.1 < \kappa < 0.9$) and RH is the relative humidity value (%). Altaratz et al. (2013) performed radiative transfer calculations using 12 years of June-August radiosonde measurements and found

that at continental stations, the AOD increased by 4% and 5% for the 1 km and 2 km layers, for $k = 0.3$, respectively, and by 5% and 4% for $k = 0.7$. That is, the effect of changes in relative humidity on AOD is limited. In this study, we have not considered the hygroscopic growth of aerosols. Assuming a 5% hygroscopic growth of AOD, the relative increase of $\frac{\partial PTT_0}{\partial AOD}$ for stratiform (convective) precipitation is 2.8% (3.3%). Such effect will not significantly change our conclusion.

And, we added a discussion of aerosol humidification effects in the revision as:

"In this study, we have not considered the aerosol humidification effect in the presence of precipitation, which may increase the retrieving error of FMF in MODIS aerosol product. Firstly, the MODIS algorithm filters out pixels within 1 km of detectable clouds, where the effect of aerosol humidification will be the greatest (Martins et al., 2002). And this algorithm significantly reduces the effect of relative humidity on aerosol optical depth retrievals (Remer et al., 2005). Secondly, Altaratz et al. (2013) performed radiative transfer calculations using 12 years of June-August radiosonde measurements and found that at continental stations, the AOD increased by 4% and 5% for the 1 km and 2 km layers, for aerosol hygroscopicity = 0.3, respectively, and by 5% and 4% for aerosol hygroscopicity = 0.7. That is, the effect of changes in relative humidity on AOD is limited. In our study, assuming a 5% hygroscopic growth of AOD, the relative increase of $\frac{\partial PTT_0}{\partial AOD}$ for stratiform (convective) precipitation is 2.8% (3.3%). Such effect will not significantly change our conclusion."

6. Lines 167-169: But it's not always the case and even rarely happens that one precipitating grid can be surrounded by eight clear-sky grids.
   A: We completely agree with the reviewer that one precipitating grid can be not always surrounded by eight clear-sky grids. Actually, such grids were excluded from this study out of this concern.

   We clarified this point in the revision as:
   "Because AOD is not available under cloudy sky, for each $1 \times 1$ grid where precipitation was detected by TRMM PR, the averaged AOD and CMAOD from the surrounding eight grids are assigned to this grid. If the AOD of all eight grids are missing, then the precipitating grids AOD were recorded as missing, and such grids were excluded from this study. Otherwise the averaged AOD from the 8 grids AOD is assigned to precipitating grid (it is not required that all 8 grids have AOD observations)."

7. Lines 169-171: Did the author take the study region as a whole when defining "dusty day"? For example, for a individual day, mean clear-sky CMAOD

surrounding precipitating grids is larger than 0.5, in this case, how did the authors deal with other clear-sky CMAOD far away from precipitation? Also classify it as dusty days? This is not clear.

A: We took the study region as a whole when defining "dusty day", and the mean CMAOD from all precipitating grids at the same day were calculated. If the mean CMAOD is larger than 0.5, then the day was defined as "dusty day". And all rain samples in that day were defined as polluted rains or dusty rains. If the mean total AOD is less than 0.2, the day was defined as pristine day, and all rain samples in that day were defined as pristine rains.

It is possible that some rainy samples in dusty days have relatively low AODs because they were far from the dust plume, and vice versa. It was found that, under this classification criteria, for convective (stratiform) precipitation, over 83% (84%) precipitating grids in pristine days showed total AOD lower than 0.2, and over 87% (79%) precipitating grids in dusty days showed CMAOD heavier than 0.5. In another word, such method can represent the main feature of aerosol condition and it has the advantage to show the large-scale atmospheric circulation as an "ensemble" comparing to the method of defining the aerosol condition for each precipitation grid separately.

We clarified this point in the revision as:
"If the mean CMAOD is larger than 0.5, then the day was defined as "dusty day". And all rain samples in that day were defined as polluted rains. If the mean total AOD is less than 0.2, the day was defined as pristine day, and all rain samples in that day was defined as pristine rains. Under this classification criteria, for convective (stratiform) precipitation, over 83% (84%) precipitating grids in pristine days showed total AOD lower than 0.2, and over 87% (79%) precipitating grids in dusty days showed CMAOD heavier than 0.5. In another word, such method can represent the main feature of aerosol condition and it has the advantage to show the large-scale atmospheric circulation as an "ensemble" comparing to the method of defining the aerosol condition for each precipitation grid separately."

8. Line 173: It's better to clarify how the authors did the spatial and temporal co-locations between TRMM and ERA5?
A: We clarified this point in the revision as:
"For each TRMM PR detected raining pixel, the daily averaged ERA5 variables averaged from all grids ±0.5° surrounded it are assigned to it."

9. Lines 227-229: It is true for convective clouds but not for stratiform clouds. Can the authors explain the reason?
A: Thanks for pointing this out. The impacts of dust aerosol on stratiform rain at low layers close to surface is weaker than that on convective rains, particularly for those stratiform rains with warmer PTTs (e.g. light blue and green curves in Fig.4d). This is because the proposed dust's IN effect generally works for ice-phase

microphysical process. For those stratiform rains starting from warm PTT, there is no sufficient water content and the temperature may be too warm for heterogeneous freezing to take place.

In the revision, we modified the statement as:
"Although followed by a layer with slower growing, the final NSRR for given PTT under dusty condition (solid curve) still is heavier than that of pristine rains (dotted curve). Such effect is weak for stratiform rains particularly those with relatively warm PTTs (e.g. light blue and green curves in Fig. 4d). This is because the proposed dust's IN effect generally works for ice-phase microphysical process. For those stratiform rains starting from warm PTTs, there is no sufficient water content and the temperatures are too warm for heterogeneous freezing to take place."

10. Line 231: Please develop a bit how dust can suppress warm rain?
A: Thanks, and we have added an explanation about this in the revision as:

"This indicates a possible suppression by dusty condition for warm rain growth. During the long-range transportation of dust from north to southeastern China, very likely the dust particles were coated by soluble aerosols and become active CCN (Li et al., 2010) in the warm rains. For given condensed liquid water content, this additional CCN leads to smaller cloud effective radius thus decreases the coalescence efficiency which is the main mechanism for warm rain growth (Rosenfeld et al., 2008; Min et al., 2009; Yin and Chen, 2007; Li et al., 2010(Li et al., 2010)."

11. Lines 236-239: As I understand, the contoured frequency by altitude diagrams is 2D probability density distribution, which represents how the data concentrate. Thus, it can not be used to illustrate if dust increases or decreases LH for a specific altitude. To do so, one should normalize data so that probability sums to 1 for each altitude, so called 'joint-histgram' .
A: Thanks for the comments. The reviewer is right, and the joint PDF can be calculated as

$$JPDF(i,j) = \frac{N(i,j)}{\sum_{i=1}^{TN1} N(i,j=j)} \times 100\%$$

where N(i,j) is the number of samples with LH in the ith bin and altitude (H) in the jth bin. TN1 is the total number of classified bins of LH. The denominator here is the total number of samples summed at certain altitude in the jth bin.

In Figure 5, we calculated the probability using total samples in the Height-LH phase space as the unified denominator.

$$PDF(i,j) = \frac{N(i,j)}{\sum_{i=1}^{TN1} \sum_{j=1}^{TN2} N(i,j)} \times 100\%$$

TN2 is the total number of bins of altitude.

Therefore, for certain altitude, the PDF(i,j) is based on the same denominator and can be compared between dusty and pristine samples.

12. Figure 6: Three LH methods are quite different with each other. I was wondering if the LH profiles are reliable? Why did author chose VPH in Figure 5? I don't see any validation studies were cited. It is expected that the results will change quite a lot and also the conclusion will not hold anymore if other two methods are used since the vertical profiles have large difference as shown in Fig. 6.

A: Thanks for the comments. Validation of satellite retrieved LH is a very challenging task (Tao et al., 2022) because there is no directly measured ground-truth of LH available. Intercomparison among different LH products is one of the useful indirect means to evaluate their accuracy. Based on Li et al., (2019), VPH product showed reasonable structure of LH in Tibetan Plateau with similarities and dissimilarities comparing to CSH and SLH.

In this study, the VPH product was chosen because it is directly related to the variations of precipitation rate at each altitude, while CSH and SLH retrievals did not use this detailed information, instead, they use constrains of precipitation rate at surface, precipitation top height, precipitation type.

Although the mean vertical profiles of LH are different among VPH, CSH and SLH, agreements are met regarding the relative difference between pristine and dusty convective rains. As shown in Figure 6, all three products agree that LH in deep convective precipitation at middle layer (around 5-6 km) in dusty condition should be stronger than those in pristine condition. For stratiform rains, VPH shows a stronger latent heat in the dusty condition near 5-6 km. There also is a slight enhancement of LH in dusty samples based on SLH and CSH products (Figure 6a, red and green curves), although this is not remarkable.

Based on the above analysis, we decided to keep the LH-related results in the manuscript, but added a discussion regarding the uncertainties of satellite LH products as this:

"Validation of satellite retrieved LH is still a very challenging task (Tao et al., 2022) because there is no directly measured ground-truth of LH available. Intercomparison among different LH products is one of the useful indirect means to evaluate their accuracy. Based on Li et al., (2019a), VPH product showed reasonable structure of LH in Tibetan Plateau with similarities and dissimilarities comparing to CSH and SLH. In this study, the VPH product was chosen because it is directly related to the variations of precipitation rate at each altitude, while CSH and SLH retrievals use constrains of precipitation rate at surface, precipitation top height, precipitation type, etc. It should be emphasized, the LH-related results did not receive rigorous validation in this study area, thus should be treated with cautions."

13. Line 254: Why the warm rain was sometimes included and but sometimes not? Any reason?

A: In section 3.3 we defined the three-layer precipitation growth rate using the method mentioned in lines 150-158 to investigate the effect of dust aerosols on the growth rate of precipitation in each layer. Those slopes include   SlopeA in the layer with temperatures colder than -5°C, SlopeB in the middle layer with temperatures between -5°C to 2°C, and SlopeC in the lowest layer with temperatures warmer than 2°C.

Because warm rain has precipitation top temperature warmer than 0 °C and there is almost no ice phase microphysical processes in it, SlopeA and SlopeB cannot be calculated from them. We have removed lines 254-255 from the text and added warm rain in the supporting information (Figs. S2 and S5).

And We added the discussion of Slope C in warm rain in the revision as:

  "As for warm rain (Fig. S2), for a given NSRR, SlopeC increases with increasing PTT. For a given PTT, SlopeC increases with NSRR. Even when both PTT and NSRR are constrained, SlopeC in dusty conditions is still significantly weaker than that in pristine conditions (Fig. S2c)."

[Figure]

Figure S2: The mean SlopeC for warm rain as functions of near surface rain rate (NSRR) and precipitation top temperature (PTT) in pristine (the left column) conditions, dusty (the middle column) conditions and the differences between them(dusty minus pristine, the right column).

"As for warm rain, the SlopeC in dusty condition is significantly smaller than that in pristine condition and the t testing showed that differences of SlopeC exceeded the 99 % confidence level (Fig. S5), indicating that dust suppressed warm rain. In addition, polluted dust particles may also act as CCN to decrease the effective radius of cloud droplets and inhibit the coalescence efficiency (warm rain) as suggested by Rosenfeld (2008), Li et al. (2010), Min et al. (2009) and Yin and Chen (2007)."

[Figure]

Figure S5: The mean Slope C as functions of precipitation top temperature (PTT) for warm rain under pristine (dotted line) and dusty (solid line) conditions (a). Overlapped are the contoured occur frequency (%) of samples under dusty conditions. For a given PTT, t test significance for the differences between SlopeC of warm rain for pristine and dusty conditions (b), red (black) line indicates the 95 % (99 %) confidence level at 100 degrees of freedom.

14. Figure 9: It's interesting that the dependence of Slope on PTT is getting stronger from C to A. Could the authors develop a bit on this? Also, Fig.9 was kind of repeating Fig.7 & 8. Although the plot types are different, all information as discussed in Fig 9 can be also seen in Fig 7&8. I recommend the author to condense a bit or put one into SI.

A: Thanks for the question.

Because the precipitation particle growth rate at upper layer (water vapor deposition process) and middle layer (aggregation and riming process) are critical to determine the final surface rain rate, SlopeA and SlopeB are more sensitive to PTT. As for SlopeC in the lower layer, the convective precipitation rate has a slight increase due to coalescence with cloud droplets. However, in the layer very close to surface, rain rate no longer grows but decreases due to breakup and/or evaporation. For the stratiform precipitation, rain rate in this layer does not grow due to the lack of updraft. Therefore, SlopeC is not sensitive to PTT.

We agree with the reviewer that Fig.9 is kind of repeating Fig.7 & 8, and we have moved Fig. 9 to the supporting information.

The above explanations were added into the revision as:

[revised manuscript text omitted]

---

## Referee Report (RR1)

The authors discuss the effects of dust aerosols and CAPE on the vertical structure of precipitation clouds using multi-source observations. This study has a certain reference significance for the study of aerosol influence uncertainty on local clouds and precipitation. The overall writing and discussion are relatively smooth. But there are still some comments that authors need to address.

1、 The authors analyzed the influence of dust aerosols on clouds and precipitation in 14 summers from 2000 to 2013. However, the probability of dust occurrence in southeast China is very low. What is the proportion of dust days in the total samples? If the proportion is too low, does the study have statistical or scientific significance?

2、 In Figure 2, the authors analyzed the backward trajectory on 12 June 2006. Please explain whether this is representative for the whole study period.

3、 How do the authors define warm rain ?

4、 In Figure 4e, why is the difference between cleaning and dust conditions minimum when the ppt is -20 to -15 (green lines)?

5、 In Figure 5, why do the negative values of the difference all appear at 5km? Does it mean that the vertical LH of the convective clouds and stratus clouds have similar feedback to dust aerosol? In addition, the color bar values displayed on the right of the figure are incomplete, please adjust them.

6、 In Figure 9, what is the criterion or basis for the selection of CAPE thresholds?

---

## Author Response (AR2)

**Answers to anonymous referee #3**

We would like to thank the reviewer for his/her time and effort reviewing our study. We have found the comments to be constructive and helpful.

In this reply, the comments from the reviewer are in black, and our answers are in red. The new text and lines of the revised document where the adjusted text can be found are also in red. In the revised document, all new text is marked in blue, and deleted text is crossed out in red.

**Minor comments:**

1. The authors analyzed the influence of dust aerosols on clouds and precipitation in 14 summers from 2000 to 2013. However, the probability of dust occurrence in southeast China is very low. What is the proportion of dust days in the total samples? If the proportion is too low, does the study have statistical or scientific significance?

   A: According to our strict definition, dusty days account for about 5% of the total precipitation days. Although the proportion of dusty days to total precipitation days is low, our sample size is still sufficient to support our study and all results were statistically significant (e.g. Figs. 9 and S9).

   Southeast China is relatively far from the original source of dust, so a relatively fixed atmospheric circulation conditions (northwest wind) is required to transport dust to this area, then under this circumstances, the observed cloud and precipitation characteristics are jointly determined by the combination of obviously different aerosol conditions and weather conditions, which creates an ideal test bed for us to isolate the combined effects from dust aerosol and meteorology conditions on precipitation. If the study area is chosen at the source of dust, its weather conditions are not uniform, it is not conducive for us to separate the two effects.

   In addition, the results from this study have significant scientific meanings. Because the purified dust aerosol effects on precipitation vertical structure should be also valid in other places, where the dynamic effects may not be isolated easily. And the method of isolating the influence of dynamical and aerosol conditions on cloud precipitation can be applied to other regions.

   We clarified this point in the Section 4 as:

   "Although the heavy dusty days in southeastern China is not frequent, the generally accompanied synoptical pattern provides us ideal testbed to isolate the dust aerosol effects from dynamic effects on precipitation. And the associated mechanism and effetcs should be also valid in other region, where the dynamic effects may not be isolated easily. And the method of isolating the influence of dynamical and aerosol

conditions on cloud precipitation can be applied to other regions."

2. In Figure 2, the authors analyzed the backward trajectory on 12 June 2006. Please explain whether this is representative for the whole study period.

A: We clarified this point in the revision:

"We likewise examined the backward trajectories of other dusty days, such as 20 June 2010 (Fig. S1), 11 June 2012 (Fig. S2), 16 June 2012 (Fig. S3), etc., the dusty air mass was from the Gobi and/or Taklamakan Desert as well. This suggests that the backward trajectory of the dusty air mass on 12 June 2006 is representative for the whole study period. Liu et al. (2011) also found that dust in southeast China originated from the Gobi Desert and Taklamakan Desert in northwestern China."

Liu, J. J., Zheng, Y. F., Li, Z. Q., Flynn, C., Welton, E. J., and Cribb, M.: Transport, vertical structure and radiative properties of dust events in southeast China determined from ground and space sensors, Atmospheric Environment, 45, 6469-6480, 10.1016/j.atmosenv.2011.04.031, 2011.

[Figure]

Figure S1: Horizontal distribution of coarse mode aerosol optical depth derived from Terra MODIS, wind fields at 500 hPa, and 72-hour back trajectories from the HYSPLIT model on 20 June 2010. Where the red box indicates the study area, the geolocation of four starting points are at 28.5° N, 111° E; 29° N, 113° E; 30° N, 110° E; and 29.5° N, 115° E, with altitudes of 1000 m (blue line), 2000 m (green line), and 4000 m (red line), extrapolated from 20 June 2010 at 13:00 UTC.

[Figure]

Figure S2: Horizontal distribution of coarse mode aerosol optical depth derived from Terra MODIS, wind fields at 500 hPa, and 72-hour back trajectories from the HYSPLIT model on 11 June 2012. Where the red box indicates the study area, the geolocation of four starting points are at 29.5° N, 115.5° E; 28.9° N, 117° E; 29.2° N, 112.5° E; and 29.5° N, 118° E, with altitudes of 1000 m (blue line), 2000 m (green line), and 4000 m (red line), extrapolated from 11 June 2012 at 18:00 UTC.

[Figure]

Figure S3: Horizontal distribution of coarse mode aerosol optical depth derived from Terra MODIS, wind fields at 500 hPa, and 72-hour back trajectories from the HYSPLIT model on 16 June 2012. Where the red box indicates the study area, the geolocation of four starting points are at 29.5° N, 110° E; 29° N, 111.5° E; 30° N, 115° E; and 28.5° N, 110° E, with altitudes of 1000 m (blue line), 2000 m (green line), and 4000 m (red line), extrapolated from 16 June 2012 at 13:00 UTC.

3. How do the authors define warm rain?

A: We clarified this point in the revision:

"Warm rain is defined as those with PTT warmer than 0 °C."

4. In Figure 4e, why is the difference between cleaning and dust conditions minimum when the ppt is -20 to -15 (green lines)?

A: When the PTT is -20°C to -15°C (green curves), the difference between cleaning and dust conditions is relatively small but still evident (as shown in Fig. 4 after excluding other curves).

[Figure]

Figure 4: Differences in vertical profiles of convective precipitation for pristine (dashed line) and dusty (solid line) conditions for given PTT. Different color stands for different PTT. Each subpanel focuses on the rain rate in the mixed layer (temperatures between -5 °C to 2 °C).

5. In Figure 5, why do the negative values of the difference all appear at 5km? Does it mean that the vertical LH of the convective clouds and stratus clouds have similar feedback to dust aerosol? In addition, the color bar values displayed on the right of the figure are incomplete, please adjust them.

A: Thanks for pointing this out. It is near the altitude of 5km that the heterogeneous freezing process dominates. The presence of dust intensifies the heterogeneous

freezing process, making it easier for ice to form, resulting in an increase in positive heating and a decrease in cooling. This process is basically the same for convective and stratiform precipitation. Therefore, the negative values of the difference (it means an increase in positive heating and a decrease in cooling) generally centered at 5km. In addition, the color bar values in Figure 5 have been adjusted, as shown below.

[Figure]

Figure 5: Contoured frequency by altitude diagrams (CFADs) of LH (retrieval from VPH) in pristine (the first column) conditions, dusty (the second column) conditions and the differences between them (the third column) for stratiform (the first row) and convective (the second row) rains.

We clarified this point in the revision as:

"Under dusty conditions, stratiform and convective rains exhibits an increased positive heating near 5 km altitude and a decrease of negative heating (cooling) at higher layer. From the difference of CFAD of LH (Fig. 5c and 5f), the negative values of the difference all appear around 5km where the heterogeneous freezing process dominates. The presence of dust intensifies the heterogeneous freezing process, making it easier for ice to form, resulting in an increase in positive heating and a decrease in cooling. The LH vertical structure of stratiform and convective rains have similar feedback to dust aerosol. Meanwhile, the cooling (i.e. negative LH) in layer lower than 5 km is also enhanced based on Fig. 5c and 5f."

6. In Figure 9, what is the criterion or basis for the selection of CAPE thresholds?

A: There are two criterions for the selection of CAPE thresholds. First, the differences between defined strong and weak CAPE groups should be great enough. Second, it is required that both groups have enough sample size. After the

experiment, it was found that the threshold of weak (strong) CAPE was taken as 25% (55%) of the cumulative probability of CAPE for pristine raining samples as more appropriate.

We clarified this point in the revision as:

"All pristine stratiform and convective raining samples are divided into two groups with strong CAPE (i.e. over 700 J kg$^{-1}$ for stratiform rains and over 1100 J kg$^{-1}$ for convective rains) and weak CAPE (i.e. weaker than 350 J kg$^{-1}$ for stratiform rains and 700 J kg$^{-1}$ for convective rains) to check the impacts of dynamic conditions on the PTT-NSRR relationship. There are two criterions for the selection of CAPE thresholds. First, the differences between defined strong and weak CAPE groups should be great enough. Second, it is required that both groups have enough sample size. After the experiment, it was found that the threshold of weak (strong) CAPE was taken as 25% (55%) of the cumulative probability of CAPE for pristine raining samples as more appropriate. "

**Answers to anonymous referee #4**

We would like to thank the reviewer for his/her time and effort reviewing our study. We have found the comments to be constructive and helpful.

In this reply, the comments from the reviewer are in black, and our answers are in red. The new text and lines of the revised document where the adjusted text can be found are also in red. In the revised document, all new text is marked in blue, and deleted text is crossed out in red.

This study investigated the impact of dust aerosol and atmospheric convective available potential energy (CAPE) on the formation of precipitating clouds in southeastern China. Overall, while the paper presents some interesting findings on the relationship between dust and CAPE and precipitation, there is one limitation that should be addressed before the potential publications.

A: We thank for the reviewer's positive comments on our work and are happy to address his/her concern as follows.

**Major comments:**

My main concern is the causality claimed in the paper. While the study provides the relationship between dust, CAPE, and the vertical structure, it is not clear to what extent these relationships are causal. For example, the impacts of dust may not be fully revealed by comparing the dust conditions and pristine conditions. As dust conditions are usually associated with certain synoptic backgrounds (i.e., strong north wind), the synoptic pattern itself will lead to the difference in CAPE. It would be helpful to see a more thorough examination of potentially confounding variables that can better isolate the effects of dust aerosols.

A: Yes, we agree that the difference in CAPE should be mainly determined by synoptic conditions instead of aerosol. CAPE, as an indicator of convection strength, makes impacts on precipitation vertical structure independently from dust aerosols. The data analysis on satellite observations confirmed that $PTT_0$ decreases 0.41-0.65ºC per 100J kg$^{-1}$ CAPE. In section 3.4, we isolated the dust aerosol effects on $PTT_0$ from CAPE effects using partial differential analysis of composite function.

Indeed, there are multiple potential confounding variables that nay lead to changes of $PTT_0$. According to the reviewer's suggestion, we conducted additional sensitivity tests of $PTT_0$ to updraft velocity (W), water vapor (RH) and wind shear in this revision using the same method for CAPE.

As shown here and in the supplementary material of Figure S12, S13 and S14, although $PTT_0$ showed some correlations to those variables (it is not surprising), the relationship of $PTT_0$ to them at 750 and 500 hPa are NOT as stable and significant as that to CAPE. For example, $PTT_0$ of stratiform (convective) precipitation shows positive (negative) correlation to updraft velocity at 750hpa but very weak correlation with it at 500hPa. Similar results were found for wind shear and water vapor. This is because the precipitation top height varied from case to case thus has different sensitivity to cloud dynamic and thermodynamic conditions at different altitudes.

Based on this analysis, CAPE as a measure of the convective instability energy has the best representativeness of dynamic effects on precipitation vertical structure.

[Figure]

Figure S12: The variation of $PTT_0$ with updraft velocity (W, Pa/s) at 500 hPa (a,c) and 750 hPa (b,d) for deep stratiform precipitation (the first row) and deep convective precipitation (the second row) under pristine conditions. The results are derived from randomly selected 70% precipitation samples from total.

[Figure]

Figure S13: As same as Figure S12, but for U and V wind shear (m/s).

[Figure]

Figure S14: As same as Figure S12, but for relative humidity (RH, %) .

In the last paragraph of Section 3.1, we emphasized that:

"It should be emphasized that the difference in CAPE should be mainly determined by synoptic conditions instead of aerosol."

In the end of Section 4 "Discussion and Conclusion", we clarified that:

"It should be noticed there are several uncertainties in this study. ……In addition, the relationship between NSRR and PTT is influenced by multiple dynamic factors. Sensitivity tests of $PTT_0$ to updraft velocity (W), water vapor (RH) and wind shear were conducted using the same method for CAPE (Figs. S12, S13 and S14). The relationship of $PTT_0$ to them at 750 and 500 hPa are not as stable and significant as that to CAPE. This is because the PTH varied from case to case and is sensitive to multiple factors at varied altitudes. CAPE as a measure of the convective instability energy has the best representativeness of dynamic effects on precipitation vertical structure. Therefore, in this study, we mainly focused on CAPE."